# Four-octyl itaconate improves osteoarthritis by enhancing autophagy in chondrocytes via PI3K/AKT/mTOR signalling pathway inhibition

Xuekang Pan[1,2,3], Huajian Shan[1,3], Jinyu Bai[1,3], Tian Gao[2], Bao Chen[2], Zhonghai Shen[2], Haibin Zhou[1], Huigen Lu [2✉], Lei Sheng [1✉] & Xiaozhong Zhou [1✉]

Osteoarthritis (OA) is a highly prevalent and chronic disorder that is associated with a substantial social and economic burden. Itaconate, as an important regulator of cellular inflammation, is a metabolite synthesised by an enzyme encoded by *immune-responsive gene 1*. However, there are few studys regarding the effects of itaconate on OA. Here, we show the effect of the cell-permeable itaconate derivative 4-octyl itaconate (OI) on OA. OI attenuates the chondrocyte apoptosis induced by interleukin 1β (IL-1β) in vitro, indicating that OI protect chondrocytes against apoptosis. Moreover, OI ameliorates the chondrocyte autophagy inhibition induced by IL-1β via the inhibition of PI3K/AKT/mTOR signalling pathway. Finally, OI enhances autophagy and reduces cartilage degradation in a rat model of OA established by destabilization of medial meniscus (DMM). In summary, our findings reveal that OI is involved in regulating the progression of OA. The above results shed light on the treatment of OA.

[1] Department of Orthopaedics, The Second Affiliated Hospital of Soochow University, 215004 Suzhou, China. [2] Department of Orthopaedics, The Second Affiliated Hospital of Jiaxing University, 314000 Jiaxing, China. [3]These authors contributed equally: Xuekang Pan, Huajian Shan, Jinyu Bai. ✉email: 13758076161@163.com; shenglei510@suda.edu.cn; zhouxz@suda.edu.cn

Osteoarthritis (OA) is a degenerative joint disorder that is highly prevalent worldwide[1]. It is mainly characterised by joint deterioration that is caused by multiple factors, including articular cartilage loss, subchondral bone remodelling and synovitis[2,3]. Late-stage OA is a major cause of disability and results in a substantial socioeconomic burden[4]. An estimated 9.6% of men and 18% of women over the age of 60 suffer from knee arthritis symptoms, and ~25% of these individuals have difficulty performing daily activities[5]. So far, there has not been any evidence that nonsurgical treatments for OA, such as nonsteroidal anti-inflammatory drugs (NSAIDs) and physiotherapy, reverse the progression of the disease[6–8]. Therefore, more effective treatments need to be studied and developed.

The imbalance of autophagy and apoptosis in chondrocytes has been shown to be the main mechanism underlying articular cartilage injury[9–11]. Autophagy is a well conserved intracellular metabolic pathway that helps maintain the homoeostasis of the intracellular environment. It strongly suppresses inflammation and is a critical component of the pathological progression of inflammatory diseases[12–15]. Articular chondrocytes activate autophagy to maintain energy metabolism, but its activity falls with age[10]. Autophagy appears to be involved in the progression of OA according to a comprehensive body of evidence[10,11,14,16–18], and promoting autophagy may be a new strategy to delay the degeneration of articular cartilage. Moreover, the PI3K/AKT/mTOR pathway plays an important role in the regulation of both cellular physiology and tumour progression[19,20], and its activation inhibits autophagy[21]. Therefore, it may be a possible target for improving autophagy by modulating the PI3K/AKT/mTOR signalling pathway and thereby improving OA[22,23].

As a metabolite, itaconate is synthesised by *immune-responsive gene 1* and contributes to the Krebs cycle[24,25]. Itaconate regulates macrophage function and inflammation[26], and it exerts a anti-inflammatory effect on activated macrophages[26]. However, its anti-inflammatory mechanism needs to be further studied[27]. Itaconate cannot penetrate the cell membrane because of its strong polarity[27], while its derivative 4-octyl itaconate (OI) is cell permeable and has chemical properties similar to those of itaconate[26], indicating that OI is a suitable substitute for itaconate and that its potential role in OA is worthy of being explored. To date, the effect of itaconate on OA remains unclear, especially its effect on autophagy. Our study looked at the effect of OI on OA and identified a part of the mechanism.

## Results

**OI protected C28/I2 cells from IL-1β-induced apoptosis**. C28/I2 cells were treated with OI (0, 10, 50, 100, 200, 300, 400, or 500 μM) for 48 h, and toxicity was assessed. OI had no significant inhibitory effect on chondrocytes at doses of 10, 50 and 100 μM, but there was a significant inhibitory effect at doses above 100 μM (Fig. 1a). Thus, we used a dose of 100 μM for subsequent analysis. The IL-1β-induced OA model is considered a classic model in vivo and in vitro[28]. Therefore, we utilised IL-1β to mimic OA in vitro. The proliferation of chondrocytes was assessed by the MTT assay (Fig. 1b). Compared with the control treatment, IL-1β treatment for 24, 48, 72 and 96 h significantly inhibited the growth of chondrocytes, but the cell growth of chondrocytes was restored after the pretreatment of OI. The IL-1β induces chondrocyte apoptosis and participates in the pathogenesis of OA[29]. Whereas, it is not clear whether OI affects chondrocyte apoptosis induced by IL-1 β. According to Fig. 1c–e, flow cytometry analyses revealed that 3.74% of the chondrocytes in the control group and 13.09% of those in the IL-1β group were apoptotic. However, IL-1β + OI treatment decreased the apoptosis rates to 8.38%. These results indicated that OI protected chondrocytes from IL-1β-induced apoptosis.

**OI induced autophagy markers and inhibited apoptosis-related factor**. Autophagy is reported to involved in the degeneration of articular cartilage, and an autophagic imbalance of chondrocytes may lead to OA[9,10]. Gene and protein expression of

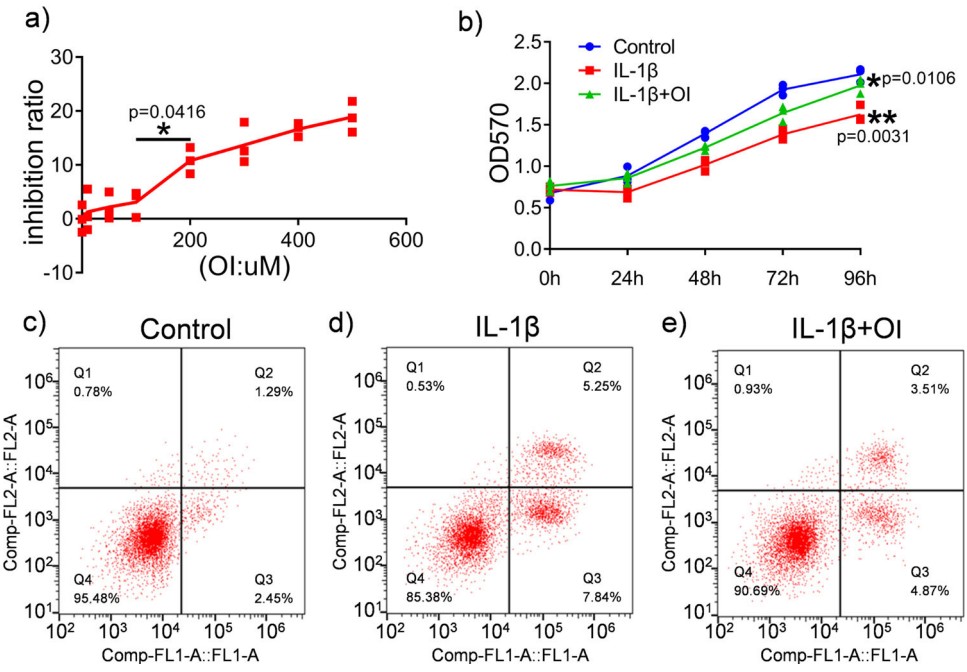

**Fig. 1 The effects of OI on C28/I2 cells. a** The cytotoxic effects of OI on C28/I2 cells was measured using the MTT assay. The highest concentration at which OI was 100 μM (n = 3). The data are the mean ± SD. *P < 0.05. **b** Cell viability after treatment with OI for 48 h before exposure to 10 ng/mL IL-1β for 24 h (n = 3). The data are the mean ± SD. *P < 0.05 IL-1β + OI group vs. the IL-1β group, **P < 0.01 IL-1β group vs. the Control group. **c–e** Flow cytometry was utilised to quantify the number of dead cells (n = 3).

autophagy-related molecules were examined to evaluate how OI affected C28/I2 cells autophagy. According to Fig. 2a–c, OI promoted the expression of the autophagy-related genes *LC3* and *Beclin1* while decreasing the expression of the *p62* gene, relative to those in the IL-1β group. Western blotting analysis of autophagy marker expression further showed that IL-1β decreased the LC3II/LC3I and Beclin-1 and increased the p62, but this effect was inhibited by OI (Fig. 2d–g). To further determine the effects of OI on the apoptosis of OA chondrocytes, the Western blotting analysis was performed. As shown in Fig. 2h, i, OI inhibited cleaved caspase-3 and BAX expression, while IL-1β promoted their expression. Together, these data suggested that OI attenuated the effect of IL-1β on the apoptosis and autophagy of chondrocytes.

**OI inhibited PI3K/AKT/mTOR signalling pathway**. The PI3K/AKT/mTOR signalling pathway is closely related to chondrocyte apoptosis and autophagy[30]. Therefore, PI3K/AKT/mTOR signalling pathway was investigated to further understand the mechanisms through which OI decreases apoptosis and promotes autophagy. We measured the protein expression of phosphorylated (p)-AKT, total (T)-AKT, phosphorylated (p)-PI3K, total (T)-PI3K, phosphorylated (p)-mTOR and total (T)-mTOR in C28/I2 cells by western blotting in different groups (Fig. 3a). In comparison to controls, PI3K, p-AKT, and p-mTOR levels were

increased in C28/I2 cells treated with IL-1β but decreased in C28/I2 cells treated with OI, which indicated that PI3K/AKT/mTOR pathway was potently inhibited by the OI treatment (Fig. 3b–d).

**OI induced autophagy marker expression by the PI3K/AKT/mTOR pathway and protected C28/I2 cells from inflammatory degradation and apoptosis**. To further explore the relationship between OI, the PI3K/AKT/mTOR signalling pathway and autophagy, we treated C28/I2 cells with chloroquine (CQ) (a specific inhibitor of autophagy) and compared the differences among the control, IL-1β, IL-1β + OI, CQ and CQ + OI groups. As it is shown in Fig. 4a, CQ slowed the proliferation of chondrocytes, while the inhibitory effect of CQ was markedly ameliorated by the addition of OI. Overall, this effect was better than that observed in the IL-1β group. Following treatment of chondrocytes with different drugs the morphology of chondrocytes was observed (Supplementary Fig. 1a–f). The chondrocytes in the control group, IL-1β + OI group and CQ + OI remained fibroblast-like and the proportion of apoptotic cells were rare. The apoptotic chondrocytes, which were indicated by shrink-age of cytoplasm and cell volume, were observed more often in the IL-1β group and CQ group. As shown in Fig. 4b, c, d and e, CQ and IL-1β blocked chondrocyte autophagy. A gene-level analysis shows that CQ and IL-1β suppress autophagy-related genes *LC3* and *Beclin1* while promoting the expression of *p62* (Fig. 4b–d).

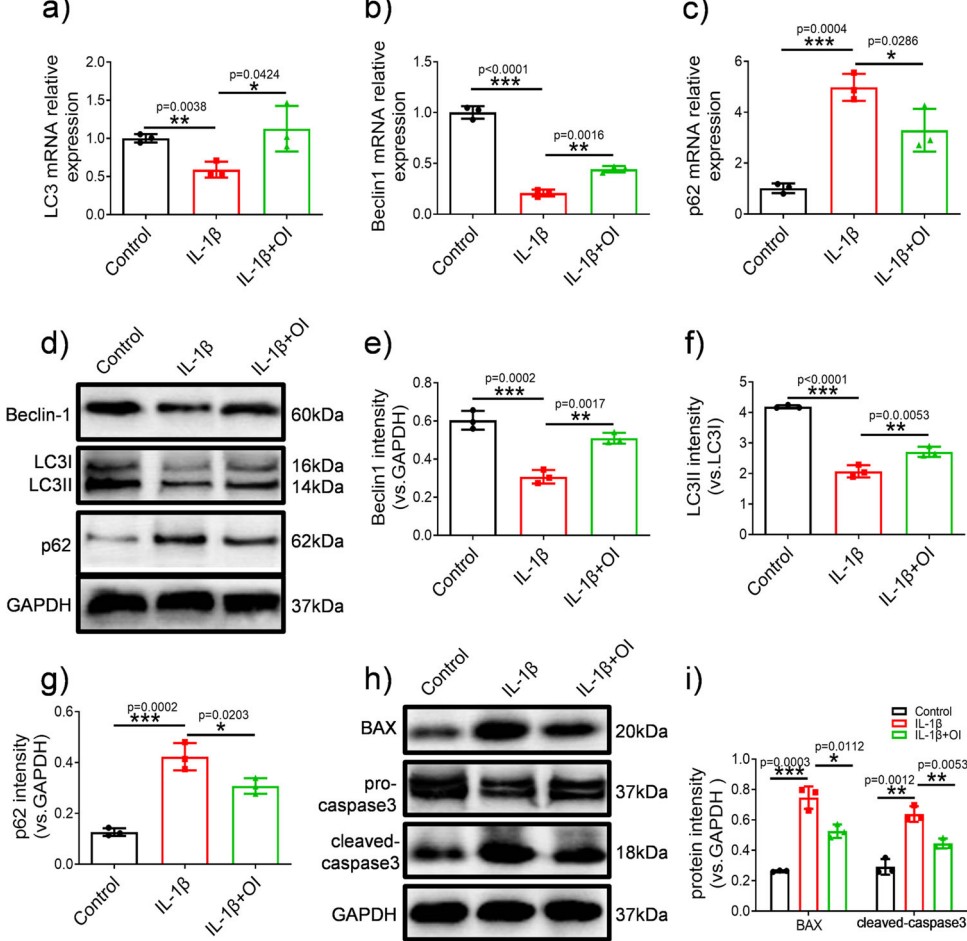

**Fig. 2 OI promoted the expression of autophagy markers and inhibited the expression of apoptosis-related proteins in C28/I2 cells. a–c** qRT-PCR was used to analyse the mRNA expression of LC3, Beclin-1, and p62 in chondrocytes treated with IL-1β and OI (*n* = 3). **d–g** WB was performed to analyse the expression of Beclin-1, LC3 and p62, which were normalised to GAPDH (*n* = 3). **h, i** Western blotting analysis of BAX, pro-caspase3 and cleaved caspase3 protein expression normalised to GAPDH (*n* = 3). Internal control was performed using GAPDH. The error bar is SD. The data are the mean ± SD.
*P < 0.05, **P < 0.01, ***P < 0.001.

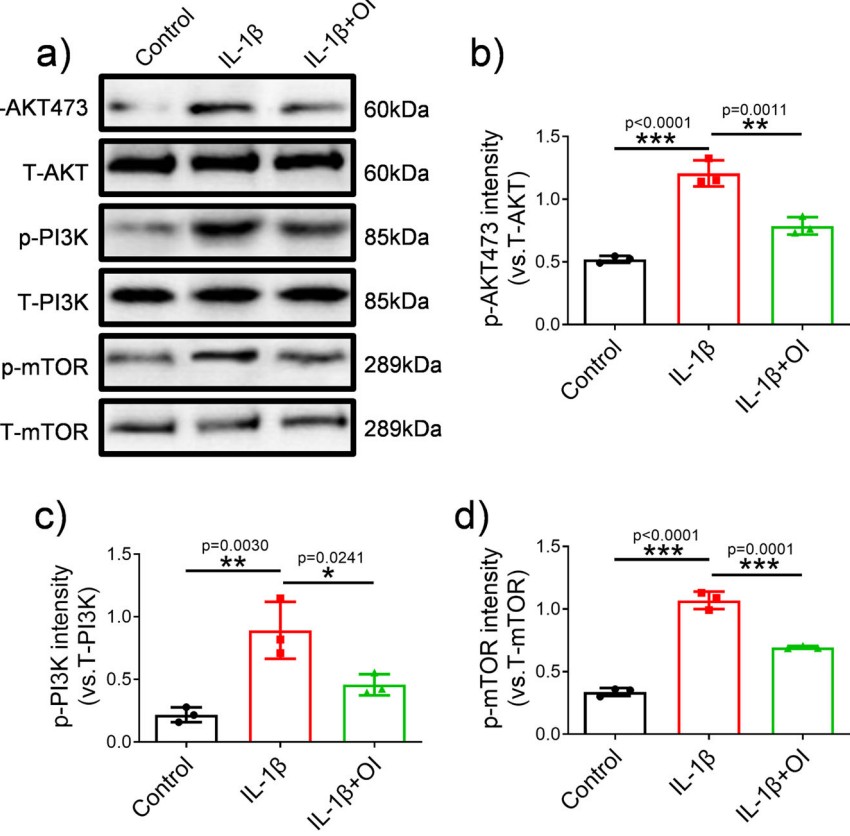

**Fig. 3 OI inhibited the PI3K/AKT/mTOR signalling pathway in C28/I2 cells. a** Representative images of Western blotting analysis of p-AKT, T-AKT, p-PI3K, T-PI3K, p-mTOR and T-mTOR in C28/I2 cells treated with IL-1β and OI ($n = 3$). **b–d** Quantification of p-AKT/T-AKT (**b**), p-PI3K/T-PI3K (**c**), and p-mTOR/T-mTOR (**d**) in the blots shown in **a**. The error bar is SD. The data are expressed by mean ± SD. *$P < 0.05$, **$P < 0.01$, ***$P < 0.001$.

Western blotting analysis (Fig. 4e–h) yielded similar results to those obtained by RT-qPCR. The expression intensity of Beclin1 in the chondrocytes of each group was measured by immunofluorescence. Figure 5d, e show that compared with CQ and IL-1β, OI significantly enhanced the fluorescence intensity of Beclin1. As such, we consider that OI attenuated the inhibitory effect of CQ and IL-1β on autophagy in chondrocytes. Western blot analyses of Fig. 4i–l showed that IL-1β and CQ markedly increased levels of phosphorylation of PI3K, AKT, and mTOR, whereas OI treatment inhibited IL-1β- and CQ-associated phosphorylation of the PI3K/AKT/mTOR signalling pathway. OI weakened the stimulating effect of IL-1β and CQ on the protein expression of cleaved caspase3 and Bax, as shown in Fig. 4m–o. To further understand the role of OI on C28/I2 cells, a gelatine zymogram was performed to detect the expression of cartilage matrix disassembly related MMP3 and MMP13 (Fig. 5a). Results showed that the gel-dissolving activity of MMP3 and MMP13 was inhibited by OI, and OI antagonised the increase in MMP3 and MMP13 expression induced by CQ and IL-1β (Fig. 5b, c).

**OI induced autophagy marker expression, reduced inflammation, inhibited apoptosis-related factor expression and improved OA in the rat model.** To evaluate the therapeutic effect of OI on rats with OA, we determined the pathological state of cartilage tissue by immunohistochemical staining, haematoxylin-eosin staining, safranin O/fast green staining, and the concentrations of inflammation-related factors in cartilage tissue were measured by the enzyme-linked immunosorbent assays (ELISAs). The immunohistochemical results in Fig. 6a–d show that in the OA model group, the expression of the cartilage degradation-related marker MMP13 was significantly enhanced

(Fig. 6a, b), while the expression of the autophagy-related marker Beclin1 was significantly decreased (Fig. 6c, d). These OA-like changes were significantly ameliorated in the rats with OA after treatment with OI (Fig. 6b, d). According to the ELISA results, as shown in Fig. 6e–h, rats treated with OI had lower serum concentrations of IL-6, TNF-α, MMP3 and MMP13 than those with OA who had not been treated. However, IL-6, TNF-α, MMP3 and MMP13 were secreted at higher levels in cartilage tissue of the rats in the OA group. As a result of Western blotting analysis, cartilage degradation-related proteins (MMP3, MMP13) and autophagy-related proteins (LC3, Beclin1, p62) were also significantly altered (Fig. 7a). The rats with OA showed dramatically attenuated transformation of LC3I to LC3II, decreased expression of Beclin1 and enhanced accumulation of p62. These phenomena were obviously altered after the injection of OI (Fig. 7f–h). OI substantially decreased the expression of apoptosis-related factors (BAX and cleaved caspase-3) in the rats with OA (Fig. 7d, e). The protein expression of related matrix metalloproteinases (MMP3 and MMP13) was downregulated by OI (Fig. 7b, c). Haematoxylin-eosin staining and safranin O/fast green staining showed that the cartilage surfaces in the rats with MTT-induced OA showed improvements after the injection of OI (Fig. 8a, b). The Osteoarthritis Research Society International (OARSI) scores also confirmed this finding (Fig. 8c), which were decreased in the OI-treated rats with destabilised medial meniscus (DMM) compared with the vehicle-treated rats with DMM, suggesting that the knee joints of the OI-treated rats exhibited fewer osteoarthritic changes than the knees of the vehicle-treated rats. These results indicated that OI could improve autophagy and reduce inflammation in rats with OA, which may have a protective effect on OA.

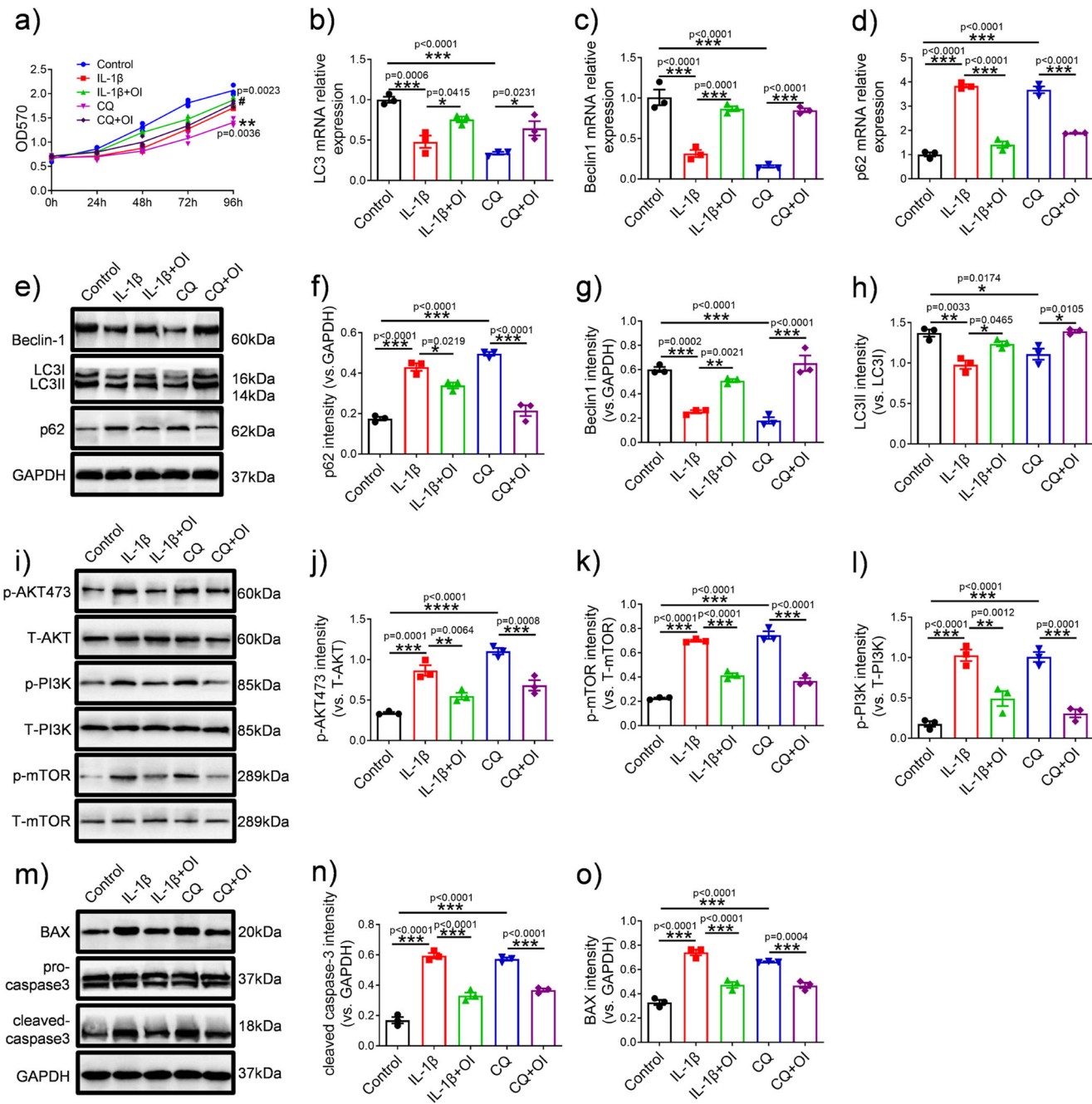

**Fig. 4 OI induced autophagy marker expression through the PI3K/AKT/mTOR signalling pathway. a** MTT assay showing the proliferation of C28/I2 cells treated with IL-1β, OI and CQ ($n = 3$). $^{\#}P = 0.0023$, the CQ + OI group vs. the CQ group, $^{**}P = 0.0036$, the CQ group vs. the Control group; **b–d** qRT-PCR was used to analyse the mRNA expression of *LC3, Beclin-1, p62* in chondrocytes treated with IL-1β, OI and CQ ($n = 3$); **e–h** WB was performed to analyse the expression levels of Beclin-1, LC3 and p62, which were normalised to GAPDH ($n = 3$); **i–l** Representative images of Western blotting analysis of the protein levels of p-AKT, T-AKT, p-PI3K, T-PI3K, p-mTOR and T-mTOR and the quantification of p-AKT/T-AKT, p-PI3K/T-PI3K, and p-mTOR/T-mTOR in the blots shown ($n = 3$); **m–o** The western blot analysis of BAX, pro-caspase3 and cleaved caspase3 protein expression and normalised to GAPDH ($n = 3$). The error bar is SD. The data are expressed by mean ± SD.$^{*}P < 0.05$, $^{**}P < 0.01$, $^{***}P < 0.001$.

## Discussion

In OA, degenerative cartilage and joint pain characterise the disease most prominently[31]. Inflammation, apoptosis, and loss of extracellular matrix play a key role in cartilage degeneration in OA[32–34]. Numerous studies have reported the cytoprotective effects of OI under different conditions. For example, OI protected human neuronal SH-SY5Y cells from hydrogen peroxide by activating the Keap1-Nrf2 signalling pathway[35]. Furthermore, OI inhibited the binding of Nrf2 to ubiquitin and enhanced the expression of Nrf2 by inhibiting E3 ubiquitin ligase (Hrd1), which inhibited the formation of osteoclasts[36]. OI regulates the metabolism of macrophages by inhibiting succinate oxidation mediated by succinate dehydrogenase, thus exerting an anti-inflammatory effect[26]. These studies indicated that OI may play a major role in cell against inflammation, and effectively maintain cell viability. The results of our study showed that chondrocyte apoptosis was reduced when chondrocytes were pretreated with OI before treatment with IL-1β, further illustrating the cytoprotective effect of OI.

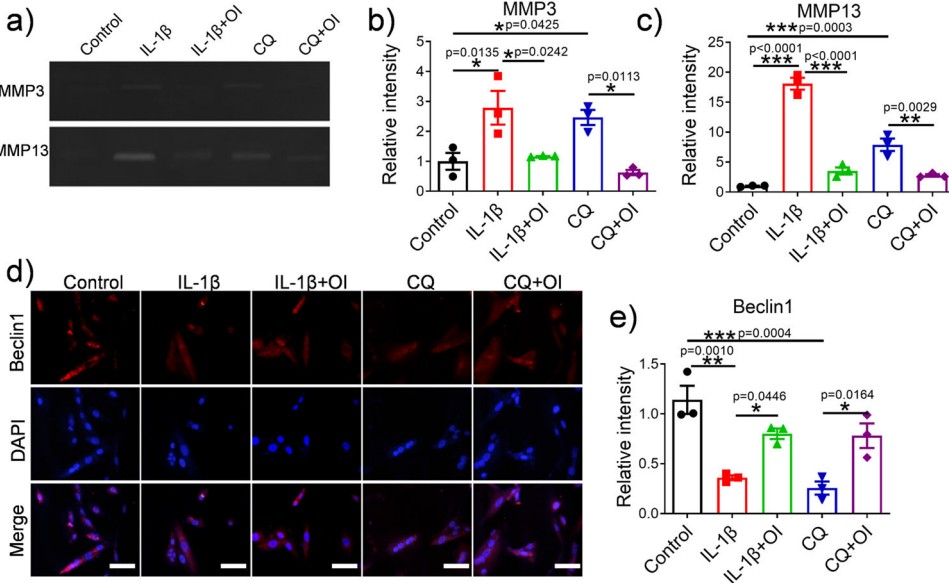

**Fig. 5 Effects of OI on the expression of cartilage degradation-related proteins and autophagy markers. a–c** Zymographic analysis of the effects of IL-1β, CQ and OI on the enzymatic activities of MMP3 and MMP13 and the quantification of MMP3 and MMP13 expression ($n = 3$). **d, e** Representative immunofluorescence photomicrograph of Beclin1 (red)-labelled chondrocytes and the fluorescence intensity quantification of Beclin1 expression ($n = 3$) (scale bar = 50 μm). Nuclei were stained with DAPI (blue). The error bar is SD. The data are expressed by mean ± SD. *$P < 0.05$, **$P < 0.01$, ***$P < 0.001$.

Furthermore, OI decreased the expression of proinflammatory proteins, including IL-6, TNF-α, MMP3 and MMP-13 in synovial fluid. Additionally, we found that OI induced apoptosis inhibition and diminished cartilage matrix degradation in OA rats, which indicates that it could improve chondrocyte survival and cartilage regeneration.

In the pathogenesis of OA, the proliferation and migration of chondrocytes are related to the dynamic balance and repair of cartilage. A critical function of autophagy is the removal of dysfunctional and damaged cells in order to maintain homoeostasis, which is closely related to the transformation of chondrocytes to osteoarthritic chondrocytes[9]. Beclin-1 and LC3 are the principal mediators involved in autophagy. In mammals, disruption of the Beclin-1/Bcl-2 complex is thought to play a critical role in activating autophagy[37]. The conversion of soluble LC3 (LC3I) to the autophagic vesicle-associated form (LC3II) is involved in autophagosome formation; thus, the ratio of LC3II/LC3I is a vital marker for autophagy[38]. When autophagy occurs, some autophagy-related proteins, such as p62, are incorporated into the mature autophagosome and degraded; as a result, p62 levels are negatively correlated with autophagy[39]. Significant inhibition of LC3II/LC3I and Beclin-1 and activation of p62 were observed in tissues isolated from an osteoarthritic animal model and human osteoarthritic chondrocytes compared with normal controls. In our prior studies, the autophagic activity of chondrocytes was increased in the early stage after IL-1β treatment, significantly inhibited after 24 h, and then remained inhibited (Supplementary Fig. 2a–d). Our findings were similar to the results of previous studies by others[18]. To eliminate the possible effects of autophagy-related compensation, we added a CQ (autophagy inhibitor)[40] group in following research. The results showed that OI activated autophagy, which decreased chondrocyte apoptosis. We found that OI strongly induced autophagy by measuring the expression of LC3II/LC3I, Beclin1 and p62. Furthermore, immunofluorescence assay for Beclin1 was performed to further ascertain that OI treatment may activate autophagy. Autophagy and apoptosis cooperate profoundly: when autophagy is activated, apoptosis-related proteins (BAX and

caspase proteins) are inhibited, whereas inhibition of autophagy can promote cell apoptosis[41,42]. In our study, we showed that IL-1β could increase apoptosis in chondrocytes, while the apoptosis of chondrocytes decreased after coincubation with OI. Therefore, we speculate that OI may inhibit the apoptosis of chondrocytes through autophagy activation.

The PI3K/AKT/mTOR signalling pathway regulates a variety of cellular processes that are essential to the maintenance of health, including cell survival, inflammation, metabolism, apoptosis and autophagy[43]. In short, PI3K/AKT/mTOR pathway is necessary for cartilage self-stabilisation[44]. mTOR is considered an important negative regulator of autophagy[45]. Therefore, we hypothesised that the regulation of autophagy in osteoarthritic chondrocytes by OI may be accomplished via the PI3K/AKT/mTOR signalling pathway. In our research, OI significantly inhibited the CQ- and IL-1β-induced activation of the PI3K/AKT/mTOR signalling pathway and apoptosis in chondrocytes. Similar results were found in our rat model of OA. The PI3K/AKT/mTOR signalling pathway is generally known to inhibit apoptosis through upstream signalling[46]. However, we discovered that caspase-3 and BAX were downregulated, and thus, inhibited the PI3K/AKT/mTOR signalling pathway. We suspected that the antiapoptotic effect of autophagy activation was more significant than the proapoptotic effect of PI3K/AKT/mTOR signalling pathway inhibition. This is worthy of further investigation. Regardless, these results suggested that OI's ability to act against OA might be mediated by mechanisms involving PI3K/AKT/mTOR signalling pathway.

In summary, our results suggest that OI may induce autophagy by regulating the PI3K/AKT/mTOR signalling pathway in chondrocytes, thereby alleviating OA-like changes. However, it remains to be determined what mechanism by which OI modulates the PI3K/AKT/mTOR signalling pathway in osteoarthritic chondrocytes. In addition, this study analysed the effects of OI primarily on chondrocytes without assessing other cell types (fibroblasts, osteoblasts), tissues (subchondral bone, meniscus), or processes. Hence, the therapeutic effect of OI on OA requires to further investigated.

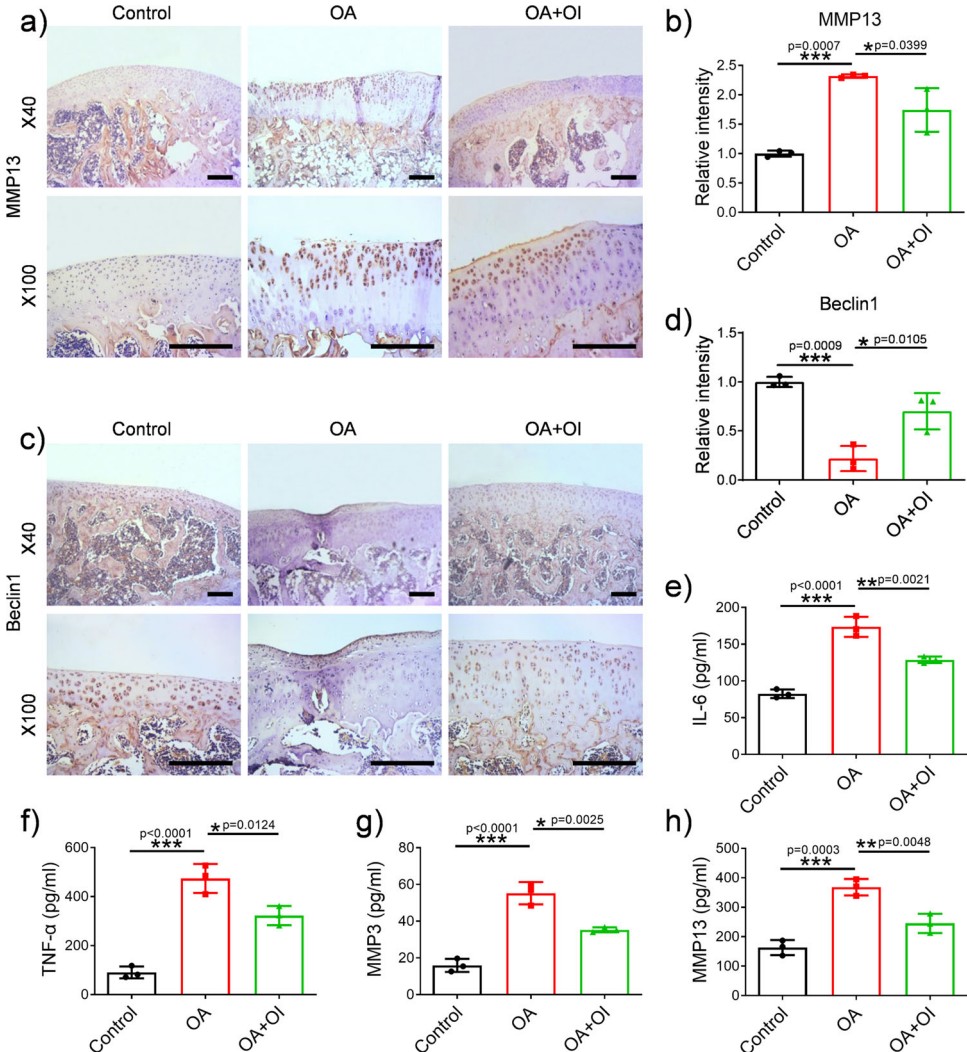

**Fig. 6 OI induced autophagy marker expression and reduced inflammatory cytokine expression in rats with OA. a–d** Representative images of immunohistochemistry and the quantification of MMP13 and Beclin-1 expression (scale bar = 250 μm). **e–h** The concentrations of IL-6, TNF-α, MMP3 and MMP13 were evaluated by ELISA ($n = 3$). The error bar is SD. The data are expressed by mean ± SD. *$P < 0.05$, **$P < 0.01$, ***$P < 0.001$.

## Methods

**Culturing and treating cells**. C28/I2 chondrocytes were obtained from HyClone (HyClone, USA). Culture of the cells were performed in DMEM/F12 containing 10% foetal bovine serum and 1% penicillin-streptomycin (HyClone, USA) at 37 °C in 5% $CO_2$. Then, the cells were stimulated with 10 ng/mL IL-1β (Beyotime, China) for 24 h. CQ (50 μM; MCE, USA) was incubated with chondrocytes for 48 h in the CQ group, chondrocytes were pretreated with OI (100 μM; MCE, USA) for 48 h followed by treatment with IL-1β (1 ng/mL) for 24 h in the IL-1β + OI group, and chondrocytes were pretreated with OI (100 μM) for 48 h followed by treatment with CQ (50 μM) for 48 h in the CQ + OI group. The control group was untreated chondrocytes cultured in regular medium.

**Cell viability assays**. We cultured C28/I2 cells in 96-well plates, and added 20 μL MTT (0.5 mg/mL) reagent (Sigma-Aldrich, USA) to each well. Then, 100 μL of DMSO (Sigma-Aldrich, USA) were filled into each well; mixing for 10 min helped dissolve the formazan dye formed by adding MTT. An Infinite M200 miniature tablet reader (Menedorf Tecan, Switzerland) was used to determine the absorbance of each sample at 490 nm.

Cell death was examined by staining C28/I2 cells with Annexin V (FITC) and propidium iodide (PI) as directed by the manufacturer. In 1.5 mL centrifuge tubes, 10 μL fluorescently labelled Annexin V reagent and 5 μL PI reagent were added, and the chondrocytes were harvested by centrifuging at $300 \times g$ for 10 min. After incubation for 10 min at room temperature in dark, around 200 μL of the solution containing cells was mixed with 2 mL of PBS in flow tubes, and these samples were analysed by flow cytometry (Thermo Fisher Scientific, USA). The assays were performed independently three times.

**Western blotting**. Cells were lysed using RIPA lysis buffer (Beyotime, China), and the protein concentrations were measured using the Bicinchoninic Acid Protein Assay Kit (Beyotime, China). The proteins were separated on a 10% SDS-polyacrylamide gel and electrotransferred to polyvinylidene difluoride (PVDF) membranes (Merck Millipore, Germany), as described previously[28]. Following 2 h of blocking with TBST supplemented with 5% non-fat dried milk, the membranes were incubated overnight with specific primary antibodies at 4 °C. Anti-PI3K (1:1000, Abcam), anti-Bcl-2 (1:1000, Abcam), anti-AKT (1:2000, Abcam), anti-pAKT (1:500, Abcam), anti-BAX (1:1000, Abcam), anti-LC3B-I (1:500, Abcam), anti-p62 (1:2000, Abcam), anti-mTOR (1:1000, Abcam), anti-cleaved caspase 3 (1:1000, Abcam), and anti-pro-caspase 3 (1:1000, Abcam) were used as primary antibodies. A next step involved washing the membranes, exposing them to peroxidase-conjugated secondary antibodies, and developing them with ELC (Amersham Pharmacia, UK); incubation at room temperature for another 60 min was performed with goat anti-rabbit IgG (1:5000) or goat anti-mouse IgG (1:5,000). Thermo provided the software and hardware for measuring immunoreactivity. The Super Signal West Femto Maximum Sensitivity Substrate Kit was used on a C-Digit Blot Scanner to measure immunoreactivity.

**Quantitative real-time polymerase chain reaction (qRT-PCR)**. Isolating total RNA from cells was accomplished using TRIzol Reagent (Invitrogen, USA). qRT-PCR was performed using the LightCycler 480 (Roche) detection system to evaluate the transcription levels. qRT-PCR was performed following the manufacturer's guidelines. GAPDH served as the internal normalisation control, and the experiments were conducted in triplicate. The primer sequences we used are listed below: p62 F: 5′-ATG AGA GAC AAA GCC AAG GAG G-3′, p62 R: 5′-CTC ACA

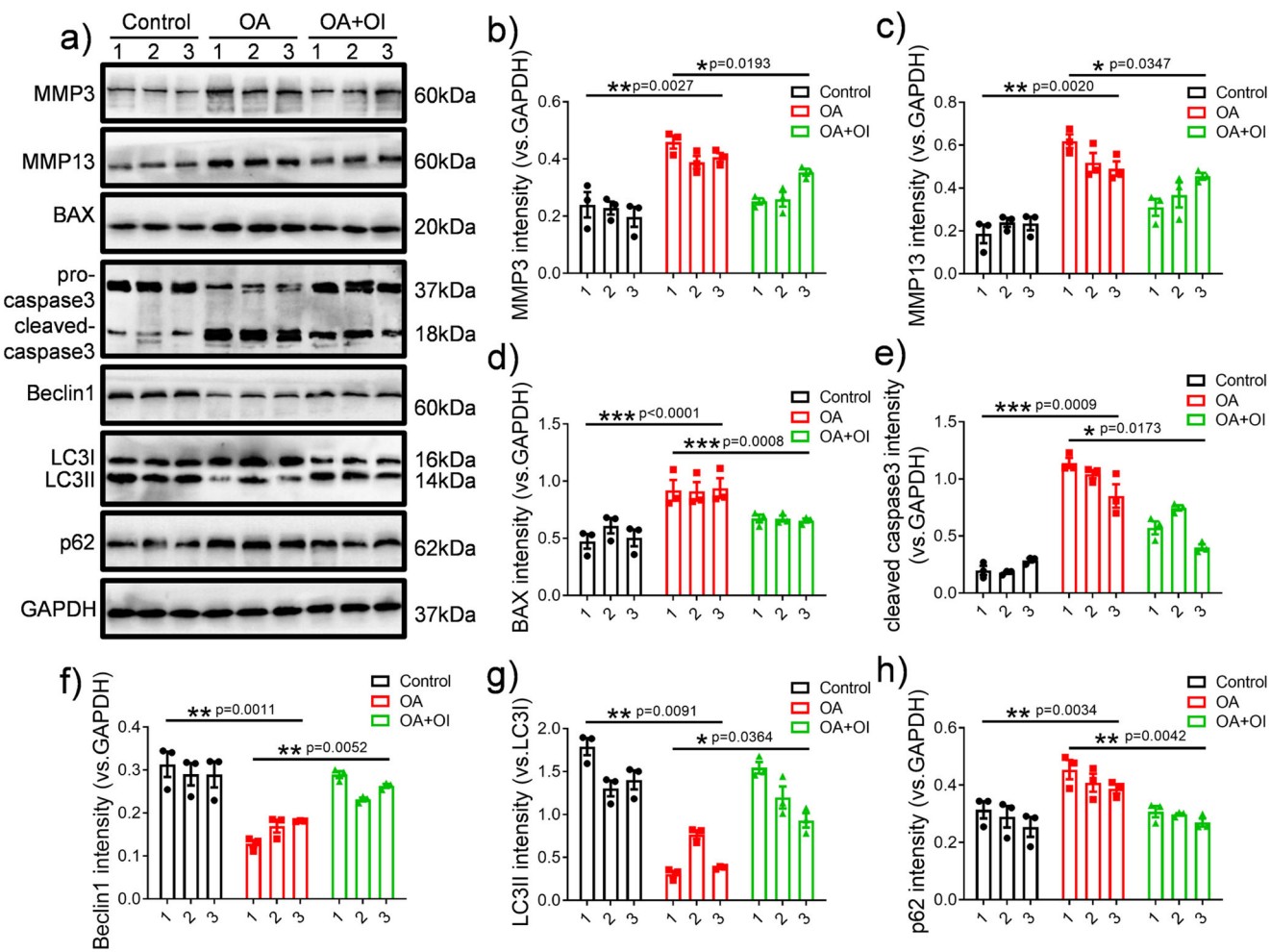

**Fig. 7 Effects of OI on the expression of cartilage degradation-related proteins, apoptosis-related proteins and autophagy-related proteins in OA rats.**
**a** The expression levels of MMP3, MMP13, BAX, pro-caspase3, cleaved caspase3, LC3, Beclin-1, p62, and GAPDH in rats were determined by western blotting and **b**–**h** normalised to that of GAPDH ($n = 3$). The error bar is SD. The data are the mean ± SD. *$P < 0.05$, **$P < 0.01$, ***$P < 0.001$.

TGG GGG TCC AAA GA-3′; LC3 F: 5′-GTC ACC GGG CGA GTT ACC-3′, LC3 R: 5′-CTT GAA AGG CCG GTC TGA GG-3′; Beclin1 F: 5′-CGA GGT GAA GAG CAT CGG G-3′, Beclin1 R: 5′-GCT GTG AGT TCC TGG ATG GT-3′; GAPDH F: 5′-AAG ACG GGC GGA GAG AAA CC-3′, GAPDH R: 5′-CGT TGA CTC CGA CCT TCA CC-3′.

**Enzyme-linked immunosorbent assay (ELISA).** The concentrations of IL-6, MMP3, MMP13 and TNF-α in the rat synovial fluid were determined by ELISA. We performed the ELISA in accordance with the manufacturer's instructions (R&D Systems, USA). The operation included adding the sample, adding enzyme, incubating, preparing the working solution, washing, dying, terminating, and detecting the results. The linear regression equation was established according to the standard concentration and OD value. Concentrations for each sample were calculated using equations, OD values, and dilution factor. The tests were carried out three times independently.

**Immunofluorescence analysis.** The chondrocytes were treated for 24 h with serum-free DMEM after reaching 80% confluence. The cells were treated according to the experimental grouping described above. Then, we washed the cells with PBS and fixed the cells for 20 min with cold methanol. For permeabilizing the fixed cells, 0.1% Triton X-100 was added for 10 min, followed by a PBS wash. Cells were then treated overnight at 4 °C in PBS supplemented with 1% BSA and an antibody against Beclin1, and Alexa Fluor® 594-conjugated secondary antibody was added and incubated for 1 h at room temperature. We stained the cell nuclei with DAPI for 5 min, mounted the coverslips with Mowiol, and observed the cells with a fluorescence microscope (Leica DM4000 B, Leica, Solms, Germany).

**Animal experiments.** Five-week-old male Sprague Dawley rats (250–300 g) were intraperitoneally injected with 2% pentobarbital sodium solution. The DMM model was used to simulate OA as previously described[47,48]. The sham controls

received a similar incision, but their ligaments were intact. During the four-week recovery period, the rats were allowed to move around freely in the cages. The rats were randomly divided into three groups: the control group (rats were sham-operated and given saline solution on the first day of every week from the 4th to the 12th week following surgery, $n = 6$), the OA group (subjected to DMM, 100 μL of normal saline treatment injected at the same as in the control group, $n = 6$), and the OA + OI group (subjected to DMM, 100 μL of OI (100 μM) injected at the same as in the control group, $n = 6$). In the end, animals were euthanized by an overdose of anaesthesia, and knee samples were harvested and fixed for at least 48 h in 4% paraformaldehyde (PFA). Cartilage and synovial fluid were harvested for further analysis. An evaluation of OA severity was conducted using the OARSI histopathology scoring system (Fig. 8c). All animal experiments were performed in accordance with the U.S. According to National Institutes of Health Guidelines (NIH Publication No. 85-23, revised 1996), the protocol was approved by Jiaxing Second Hospital Ethics Committee (JXEY-2020JX051).

**Rat articular cartilage sample preparation and immunohistochemical analysis.** Rat knee joints were fixed with 4% PFA and decalcified in 10% EDTA solution. This step was followed by alcohol dehydration and paraffin embedding. Paraffin sections were cut to 5-μm thicknesses, subsequently deparaffinized with xylene, and hydrated using a gradient. Immunohistochemistry, haematoxylin-eosin staining and safranin O/fast green staining were performed using standard protocols.

**Statistical analysis and reproducibility.** Experimental data was analysed with SPSS 17.0 software, and the results are presented as mean ± standard deviation (mean ± SD). A *t*-test was used to analyse the difference between the two groups. Single-factor analysis of variance was used for comparisons among groups (one-way ANOVA). The data is formatted with GraphPad Prism Version 5.0 software. The experiments were conducted independently at least three times with duplicate samples. Differences were statistically significant ($P < 0.05$). *$P < 0.05$; **$P < 0.01$; ***$P < 0.001$; ns, no significance.

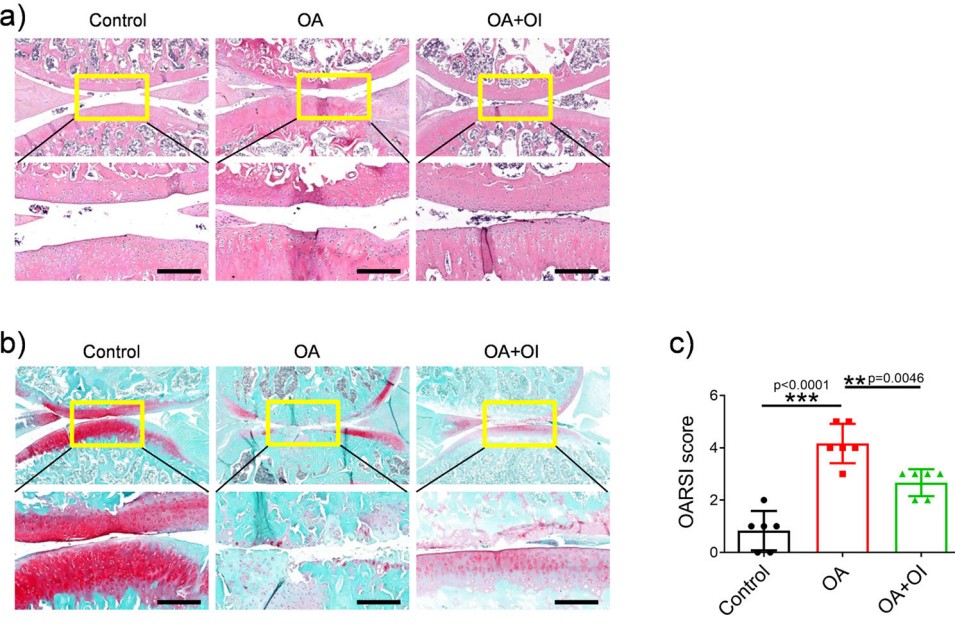

**Fig. 8 OI improved osteoarthritis in rats.** Histological analysis of knees from Control-, AO- and AO + OI-treated rats. The sections of knee joints were stained with **a** haematoxylin-eosin staining and **b** safranin O/fast green staining after 8 weeks of treatment. **c** OARSI scores were computed using safranin O/fast green stain. Scale bar = 200 μm. The error bar is SD. The data are the mean ± SD (n = 6). **P < 0.01, ***P < 0.001.

**Reporting summary**. Further information on research design is available in the Nature Research Reporting Summary linked to this article.

## Data availability

The datasets generated and/or analysed during the current study can be found in Supplementary Data. 1 or are available from the corresponding author upon request.

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

## Acknowledgements
The study was supported by the National Natural Science Foundation of China (81873995), Social Development Key Programs of Jiangsu Province-Advanced Clinical Technology (BE2019662, BE2018656), Scientific Research Foundation of Jiangsu Provincial Health and Family Planning Commission (H2017066).

## Author contributions
H.G.L., L.S. and X.Z.Z. designed and supervised the project. X.K.P., H.J.S. and J.Y.B. performed the experiments and collected the data. X.K.P., H.J.S. and J.Y.B. contributed equally to this manuscript. X.K.P., B.C., T.G., Z.H.S. and H.B.Z. analysed and interpreted the data, contributed to the writing of the manuscript, discussed the results and implications, and edited the manuscript.

## Competing interests
The authors declare no competing interests.

## Additional information

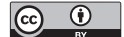

