## [Peer Review File · Communications Biology]

Reviewers' comments:

Reviewer #1 (Remarks to the Author):

The manuscript from Pan et al. describes the effect of cell permeable 4-octyl itaconate 4-OI on the human chondrocyte cell line C28/I2, and on an in vivo rat model of post-traumatic OA. The authors show that 4-OI reduces cell apoptosis and rescues proliferation of C28/I2 cells in response to IL-1 β . Next, the authors show that markers of autophagy are increased while markers of autophagy inhibition are decreased. Furthermore, the authors show that 4-OI inhibited increases in PI3K/AKT/mTOR signaling in vitro. Next, the authors show that co-incubation of 4-OI with chloroquine, an autophagy inhibitor, reverses the anti-apoptotic and molecular effects seen by 4-OI. Finally, the authors inject 4-OI into joints of rats that received surgery to induce post-traumatic OA. The authors show that Beclin-1 is rescued and MMP-13 is reduced in vivo. Furthermore, markers of autophagy, apoptosis, cartilage degeneration and PI3K/AKT/mTOR signaling were modified in cartilage by 4-OI injection in vivo while circulating inflammatory markers were reduced.

Overall, the manuscript contains some interesting data indicating potential disease modifying effects of 4-OI for treatment of OA. However, there are some significant issues with the manuscript related to missing data and statistical analyses, conclusions being overstated, figure resolution and organization, and writing clarifications. Details of these issues are indicated below:

1) Line 69 – Can the authors clarify in the introduction what a mercaptan reaction is. This term should be defined and introduced here.

2) Throughout the results it is unclear, specifically for experiments using C28/I2 cells, how many independent experiments were performed and the number of technical replicates to create the mean from each independent experiment were used. Furthermore, for animal experiments, how many animals were used for each analysis? It is recommended that the authors change the data from plunger plots to dot plots to specifically detail each independent experiment/in vivo measure.

3) Line 80 – “moi” should be “M”

4) In a number of instances in the manuscript, when the authors discuss proliferation and apoptosis, there is no statistical analyses associated with these experiments, however, the authors indicate in the results writing that apoptosis was significantly inhibited and proliferation was rescued. Statistical analyses are needed to back-up these results. This includes All of figure 1 and panels in figure 4.

5) Line 92 – The title of this section is recommended to be amended. The data in Figure 2 only shows markers, not actual activity of autophagy and apoptosis, thus the section title should be changed accordingly.

6) In figure 2C, the pro-caspase-3 blot appears to be over saturated. Do the authors have a lower exposure of this blot to see differences in expression better? Similar situations appear to be the case in figures 4C&E.

7) Figure 4 and lines 124-153 - Figure 4 is completely out of order compared to what is described in the results. In addition, the blots and graphs in figure 4 are difficult to see as they are very small and cannot be accurately reviewed at this resolution.

8) Short-forms such as “p”, “t” and “CQ” should be listed in their long form at first instance in the manuscript.

9) Line 130 – can the authors clarify where the apoptosis data is in figure 4A? The data presented looks to be proliferation data.

- 10) Line 131 – the proliferation data in figure 4 needs statistical analysis.
- 11) Figure 4G – this data would benefit from some quantification as the authors indicate on line 141 that the fluorescence intensity was significantly enhanced, but without statistics to back this up.
- 12) Can the authors clarify if the zymogram activity was quantified in figure 4F? This needs to be done for appropriate statistical analysis.
- 13) In the methods, the authors describe their surgery as a DMM model. The surgery is actually a meniscectomy and not a DMM. Furthermore, the authors have not shown that the model induced OA, particularly cartilage degeneration. OARSI scoring or similar scoring of the cartilage is needed to confirm OA was indeed induced in the model.
- 14) Figure 5A and lines 160-165 - These statements require quantification for MMP13 and Beclin. It is recommended that positive cells/total cells be calculated.
- 15) Lines 173-175 – Can the authors clarify where secretion data from cartilage tissue can be found? There is measures of these markers in the serum but secretion data from cartilage is missing.
- 16) Lines 194-196 – The data presented for in vitro studies is prophylactically cytoprotective. However, the in vivo results suggest that 4-OI attenuates disease after induction OA. What happens in culture when cells are treated with IL-1B first and followed by 4-OI? This would be a good experiment to support the in vivo finding using in vitro means to show that 4-OI could have therapeutic potential.
- 17) Line 206 - Chondrocytes do not transform to OA. It is recommended that the authors revise this statement.
- 18) Lines 222-224 – Can the authors clarify how fluorescence intensity of Beclin-1 more directly proved the ability of 4-OI to activate autophagy of chondrocytes?
- 19) Lines 243 - Although the molecular results are of interest, more attention needs to be paid to the in vivo results, as these results have more translatable potential vs. the in vitro results provided by the authors.

Reviewer #2 (Remarks to the Author):

This paper mainly investigated the effects of 4-octyl itaconate, a cell-permeable itaconate derivative(OI) on OA in vivo and in vitro.This study has some useful values.

In addition, there are some flaws in the manuscript and they should be improved.

1) The experiments in vivo seems not enough. Whether the protective effects of OI on cartilages should be evaluated such as safranin O staining and hematoxylin and eosin staining, etc.

2)All figures seems not clearly identified. X-axis and Y-axis. Actually the font sizes are too small.

3)The manuscript contains some grammar errors and imprecise expression and need careful editing by native speaker before resubmission.

Responses to reviewer's comments

We sincerely thank the reviewers for their constructive comments and suggestions. We have revised the manuscript accordingly and detailed the point-by-point response below.

1. **The reviewer's comment:** Line 69 – Can the authors clarify in the introduction what a mercaptan reaction is. This term should be defined and introduced here.

69 (OI)[24]. OI and Itaconate have similar activity of mercaptan reaction, which makes
70 OI a suitable substitute for itaconate to study biological function. Itaconate has a

Responses: Thank for your professional review on our manuscript. The “mercaptan reaction” is a kind of chemical reaction and is of little concerns with this study. The description is deleted. (Page.3, Line 66-67)

2. **The reviewer's comment:** Throughout the results it is unclear, specifically for experiments using C28/I2 cells, how many independent experiments were performed and the number of technical replicates to create the mean from each independent experiment were used. Furthermore, for animal experiments, how many animals were used for each analysis? It is recommended that the authors change the data from plunger plots to dot plots to specifically detail each independent experiment/in vivo measure.

Responses: Thank you very much for your comments. We changed the data from plunger plots to dot plots to specifically detail each independent experiment/in vivo

measure. For details, please check the figure section. (Fig.1-Fig.8)

3. **The reviewer's comment:** Line 80 – “moi” should be “M”.

Responses: We are very sorry to have such a problem, we have revised accordingly. (Page.3, Line76)

4. **The reviewer's comment:** In a number of instances in the manuscript, when the authors discuss proliferation and apoptosis, there is no statistical analyses associated with these experiments, however, the authors indicate in the results writing that apoptosis was significantly inhibited and proliferation was rescued. Statistical analyses are needed to back-up these results. This includes All of figure 1 and panels in figure 4.

Responses: Thank you very much for your comments. We have added the statistical analysis of Figure 1 and Figure 4. In addition, statistical analysis of all data was performed. (Fig.1-Fig.8).

Fig. 1 The effects of OI on C28/I2 cells **a)** The cytotoxic effects of OI on C28/I2 cells was measured using the MTT assay. The highest concentration at which OI was 100 μ M (n=3). The data are the mean \pm SD. *, P<0.05. **b)** Cell viability after treatment with OI for 48 h before exposure to 10 ng/mL IL-1 β for 24 h (n=3). The data are the mean \pm SD. *, P<0.05 vs. the Control group, #, P<0.05 vs. the IL-1 β group. **c-e)** Flow cytometry assay was performed to assess the number of dead cells (n=3).

Fig. 4 OI induced autophagy marker expression through the PI3K/AKT/mTOR signaling pathway. **a)** MTT assay showing the proliferation of C28/I2 cells treated with IL-1 β , OI and CQ (n=3). *, P<0.05, the CQ+OI group vs. the CQ group, the CQ group vs. the Control group; **b-d)** qRT-PCR was used to analyse the mRNA expression of *LC3*, *Beclin-1*, *p62* in chondrocytes treated with IL-1 β , OI and CQ (n=3); **e-h)** WB was performed to analyse the expression levels of Beclin-1, LC3 and p62, which were normalized to GAPDH (n=3); **i-l)** Representative images of Western blotting analysis of the protein levels of p-AKT, T-AKT, p-PI3K, T-PI3K, p-mTOR and T-mTOR and the quantification of p-AKT/T-AKT, p-PI3K/T-PI3K, and p-mTOR/T-mTOR in the blots shown (n=3); **m-o)** The Western blot analysis of BAX, pro-caspase3 and cleaved-caspase3 protein expression and normalized to GAPDH (n=3). The data are expressed by mean \pm SD. *, P<0.05, **, P<0.01, ***, P<0.001.

5. ***The reviewer's comment:*** Line 92 – *The title of this section is recommended to be amended. The data in Figure 2 only shows markers, not actual activity of autophagy and apoptosis, thus the section title should be changed accordingly.*

Responses: I am very grateful to your comments for the manuscript. In order to express the view of the article more accurately, we changed the section title to “OI promoted the expression of autophagy markers and inhibited the expression of apoptosis-related factors in C28/I2 cells”.(Page.4, Line 91-92) At the same time, we have also changed the expression of other relevant parts of the article. (Page.5, Line 118-120, Line 143-144)

6. ***The reviewer's comment:*** *In figure 2C, the pro-caspase-3 blot appears to be over saturated. Do the authors have a lower exposure of this blot to see differences in expression better? Similar situations appear to be the case in figures 4C&E.*

Responses: Thanks for your reminder. The problem with the expression intensity of WB may be that our laboratory all used a slightly high intensity exposure when doing WB, which caused the WB results to be slightly over-exposed, here, we added a quantitative data to better understand the trend of proteins expression (Fig.1-Fig.8).

7. ***The reviewer's comment:*** Figure 4 and lines 124-153 - Figure 4 is completely out of order compared to what is described in the results. In addition, the blots and graphs in figure 4 are difficult to see as they are very small and cannot be accurately reviewed at this resolution.

Responses: We acknowledge your comments and suggestions very much, which are valuable in improving the quality of our manuscript. We adjusted and split the figure 4 and changed the description in the results in order. (Page.5, Line120-141, Figure 4-5) The details of the changes are as follows:

Fig. 4 OI induced autophagy marker expression through the PI3K/AKT/mTOR signaling pathway. **a)** MTT assay showing the proliferation of C28/I2 cells treated with IL-1 β , OI and CQ (n=3). *, P<0.05, the CQ+OI group vs. the CQ group, the CQ group vs. the Control group; **b-d)** qRT-PCR was used to analyse the mRNA expression of *LC3*, *Beclin-1*, *p62* in chondrocytes treated with IL-1 β , OI and CQ (n=3); **e-h)** WB was performed to analyse the expression levels of Beclin-1, LC3 and p62, which were normalized to GAPDH (n=3); **i-l)** Representative images of Western blotting analysis of the protein levels of p-AKT, T-AKT, p-PI3K, T-PI3K, p-mTOR and T-mTOR and the quantification of p-AKT/T-AKT, p-PI3K/T-PI3K, and p-mTOR/T-mTOR in the blots shown (n=3); **m-o)** The Western blot analysis of BAX, pro-caspase3 and cleaved-caspase3 protein expression and normalized to GAPDH (n=3). The data are expressed by mean \pm SD. *, P<0.05, **, P<0.01, ***, P<0.001.

Fig. 5 Effects of OI on the expression of cartilage degradation-related proteins and autophagy markers. a-c) Zymographic analysis of the effects of IL-1 β , CQ and OI on the enzymatic activities of MMP3 and MMP13 and the quantification of MMP3 and MMP13 expression (n=3). **d-e)** Representative immunofluorescence photomicrograph of Beclin1 (red)-labelled chondrocytes and the fluorescence intensity quantification of Beclin1 expression (n=3) (scale bar = 50 μ m). Nuclei were stained with DAPI (blue). The data are expressed by mean \pm SD. *, P<0.05, **, P<0.01, ***, P<0.001.

8. *The reviewer's comment: Short-forms such as "p", "t" and "CQ" should be listed in their long form at first instance in the manuscript.*

Responses: Thank you very much for your comments. We have modified it accordingly. (Page.4, Line 110-112, Page.5, Line121). The details of the changes are as follows:

"Western blotting was performed to measure the protein expression of phosphorylated (p)-AKT, total (T)-AKT, phosphorylated (p)-PI3K, total (T)-PI3K, phosphorylated (p)-mTOR and total (T)-mTOR in C28/I2 cells in different groups (Fig. 3a)."

“we treated C28/I2 cells with chloroquine (CQ) (a specific inhibitor of autophagy) and compared the differences among the control, IL-1 β , IL-1 β +OI, CQ and CQ+OI groups.”

9. **The reviewer’s comment:** Line 130 – can the authors clarify where the apoptosis data is in figure 4A? The data presented looks to be proliferation data.

130 in Fig. 4A, CQ accelerated the apoptosis of chondrocytes, while the chondrocytes
131 growth inhibition effect of CQ was also significantly improved with the addition of
132 OI. Overall, this effect was better than that of the IL-1 β group. This suggested that OI

Responses: Thank you very much for your reminding. The data in figure 4A is real proliferation data. We made a writing error in previous version of the manuscript. The details of the changes are as follows:

“As shown in Fig. 4a, CQ slowed the proliferation of chondrocytes, while the inhibitory effect of CQ was markedly ameliorated by the addition of OI.” (Page.5, Line 123-125)

10. **The reviewer’s comment:** Line 131 – the proliferation data in figure 4 needs statistical analysis.

Responses: We acknowledge your suggestion very much. We supplement the statistical analysis in figure 4a.

Fig.4a: MTT assay of the proliferation rate of chondrocytes treated with IL-1 β , OI and CQ (n=3). *, P<0.05 CQ+OI group vs CQ group, CQ group vs Control group.

11-12. The reviewer's comment: 11. Figure 4G – this data would benefit from some quantification as the authors indicate on line 141 that the fluorescence intensity was significantly enhanced, but without statistics to back this up.

12. Can the authors clarify if the zymogram activity was quantified in figure 4F? This needs to be done for appropriate statistical analysis.

Responses: We sincerely appreciate the valuable suggestions of the reviewer. The data of immunofluorescence and Zymographic were quantitatively and statistically analyzed.

Fig. 5 Effects of OI on the expression of cartilage degradation-related proteins and autophagy markers. a-c) Zymographic analysis of the effects of IL-1 β , CQ and OI on the enzymatic activities of MMP3 and MMP13 and the quantification of MMP3 and MMP13 expression (n=3). **d-e)** Representative immunofluorescence photomicrograph of Beclin1 (red)-labelled chondrocytes and the fluorescence intensity quantification of Beclin1 expression (n=3) (scale bar = 50 μ m). Nuclei were stained with DAPI (blue). The data are expressed by mean \pm SD. *, P<0.05, **, P<0.01,

***, $P < 0.001$.

13. The reviewer's comment: *In the methods, the authors describe their surgery as a DMM model. The surgery is actually a meniscectomy and not a DMM. Furthermore, the authors have not shown that the model induced OA, particularly cartilage degeneration. OARSI scoring or similar scoring of the cartilage is needed to confirm OA was indeed induced in the model.*

Responses: We acknowledge the your comments and suggestions very much, which are valuable in improving the quality of our manuscript. We restated the surgical methods and added related references. (Page.12, Line 320-334) The OARSI scoring was employed to confirm the OA model. (Page.12, Line 333-334) The details of the changes are as follows:

“In this experiment, 5-week-old male Sprague Dawley rats (250–300 g) were anaesthetized with 2% pentobarbital sodium solution via intraperitoneal injection. The DMM model was used to simulate OA as previously described^{48, 49}. For the sham- operated controls, a similar incision was made on another leg of the rats, but the ligaments were left intact. The rats were allowed to recover for 4 weeks, and during this time they were allowed to move freely in the cages. The rats were randomly divided into three groups: the control group (sham- operated and treated with 100 μ L of normal saline on the first day of every week from the 4th to 12th week after surgery, $n=6$), the OA group (subjected to DMM, 100 μ L of normal saline treatment injected at the same as in the control group, $n=6$), and the OA+OI group (subjected to DMM, 100 μ L of OI (100 μ M) injected at the same as in the control group, $n=6$). At the end of the treatments, the animals were euthanized by an overdose of anaesthesia, and knee samples were harvested and fixed with 4% paraformaldehyde (PFA) for at least 48 h. Cartilage and synovial fluid were harvested for further analysis. The OARSI histopathology scoring system was used to evaluate OA severity (Fig. 8c).”

14. The reviewer's comment: *Figure 5A and lines 160-165 - These statements require*

quantification for MMP13 and Beclin1. It is recommended that positive cells/total cells be calculated.

Responses: We acknowledge your suggestion very much. We consider that MMP13 and Beclin1 are expressed more or less in the joint tissue of rats, so we used the intensity of expression for quantitative analysis. The details of the changes are as follows:

Fig. 6 OI induced autophagy marker expression and reduced inflammatory cytokine expression in rats with OA. **a-d)** Representative images of immunohistochemistry and the quantification of MMP13 and Beclin-1 expression (scale bar = 250-100 μ m). **e-h)** The concentrations of IL-6, TNF- α , MMP3 and

MMP13 were evaluated by ELISA (n=3). The data are expressed by mean \pm SD. *, P<0.05, **, P<0.01, ***, P<0.001.

15. The reviewer's comment: Lines 173-175 – *Can the authors clarify where secretion data from cartilage tissue can be found? There is measures of these markers in the serum but secretion data from cartilage is missing.*

Responses: We sincerely appreciate the valuable suggestions. I'm sorry for our unclear description. Actually, the synovial fluid of rats was harvested for ELISA experiment. We have changed the description in the article. (Page.11, Line 301-302) The details of the changes are as follows:

“The concentrations of IL-6, MMP3, MMP13 and TNF- α in the rat synovial fluid were determined by ELISA.”

16. The reviewer's comment: Lines 194-196 – *The data presented for in vitro studies is prophylactically cytoprotective. However, the in vivo results suggest that 4-OI attenuates disease after induction OA. What happens in culture when cells are treated with IL-1B first and followed by 4-OI? This would be a good experiment to support the in vivo finding using in vitro means to show that 4-OI could have therapeutic potential.*

194 our study showed that the apoptosis of chondrocytes was reduced when chondrocytes
195 were pre-treated with OI before the addition of IL-1 β , which is another example of the
196 cytoprotective effect of OI. In our results, the toxic effect of different doses of OI on

Responses: We acknowledge the reviewer's comments and suggestions very much. The inflammatory degeneration of chondrocytes stimulated by IL-1 β is rapid, in order to avoid early rapid apoptosis caused by sudden stimulation of chondrocytes, we chose OI pretreatment. We believe that it can better reflect the real role of OI in chondrocytes. Similar behavior has been reported in literatures¹⁻³.

17. The reviewer's comment: Line 206 - *Chondrocytes do not transform to OA. It is recommended that the authors revise this statement.*

Responses: Thank you for reminding this mistake. We have revised accordingly. (Page.7, Line 198-199) The details of the changes are as follows:

“which is closely related to the transformation of chondrocytes to OA chondrocytes.”

18. The reviewer’s comment: Lines 222-224 – Can the authors clarify how fluorescence intensity of Beclin-1 more directly proved the ability of 4-OI to activate autophagy of chondrocytes?

Responses: We sincerely appreciate the valuable suggestions of the reviewer. Our original manuscript did not accurately express the difference between autophagy markers and autophagy, we have corrected a number of inappropriate expressions on autophagy. (Page.4, Line 91-92; Page.5, Line 143-144, Line 130-132)

19. The reviewer’s comment: Lines 243 - Although the molecular results are of interest, more attention needs to be paid to the in vivo results, as these results have more translatable potential vs. the in vitro results provided by the authors.

Responses: We acknowledge your suggestion very much. In the animal experiment part, we added relevant animal experiments to further illustrate the role of OI on alleviating OA. The details of the addition are as follows:

Fig. 8 OI improved osteoarthritis in OA rats. Histology of Control, AO and AO+OI

rat knee model. The sections of knee joints were stained by **a)** hematoxylin- eosin staining (scale bar = 50-100 μm) and **b)** safranin O/fast green staining (scale bar = 50-100 μm) after eight weeks of treatment. **c)** The OARSI score was evaluated based on safranin O/fast green staining. Data are means \pm SD (n=6). **, P<0.01, ****, P<0.0001.

About language and grammar: We carefully examined the details of the article, corrected the errors in the article, and modified the format according to the requirements of the journal.

References

1. Hua, L., F. Yu, X. Min, Y. Jian, G. Di, Four-octyl itaconate activates Keap1-Nrf2 signaling to protect neuronal cells from hydrogen peroxide. *Cell Communication and Signaling*, 2018. 16(1).
2. Linghu, K.-G., S.H. Xiong, G.D. Zhao, T. Zhang, W. Xiong, M. Zhao, X.-C. Shen, W. Xu, Z. Bian, Y. Wang, H. Yu, *Sigesbeckia orientalis* L. Extract Alleviated the Collagen Type II-Induced Arthritis Through Inhibiting Multi-Target-Mediated Synovial Hyperplasia and Inflammation. *Frontiers in pharmacology*, 2020. 11: p. 547913-547913.
3. Ohta, M., N. Chosa, S. Kyakumoto, S. Yokota, N. Okubo, A. Nemoto, M. Kamo, S. Joh, K. Satoh, A. Ishisaki, IL-1 β and TNF- α suppress TGF- β -promoted NGF expression in periodontal ligament-derived fibroblasts through inactivation of TGF- β -induced Smad2/3- and p38 MAPK-mediated signals. *International journal of molecular medicine*, 2018. 42(3): p. 1484-1494.

Reviewers' comments:

Reviewer #2 (Remarks to the Author):

This paper mainly investigated the effects of 4-octyl itaconate, a cell permeable itaconate derivative on the efficacy osteoarthritis in C28I2 cells and DMM rat models. The authors found that OI can alleviate OA through inhibiting PI3K/AKT/mTOR pathway. This study has some academic values. The revised manuscript seems improved.

1. In Fig.4, when the authors discussed OI effects on apoptosis and autophagy, the cell morphology changes are suggested to be described at the same time since they are intuitive, not only the cellular viability .

2. In Fig.6, the magnification times should be total objective magnification times, that is magnification times * eyepiece magnification times, not objective magnification.

3. The concentrations of IL-6, TNF- α should be easily and more accurately assayed and in the serum more in the synovial fluid. MMP3 and MMP13 may be detected using immunohistochemistry.

Responses to reviewer's comments

We sincerely thank the reviewers for their constructive comments and suggestions. We have revised the manuscript accordingly and detailed the point-by-point response below.

1. *The reviewer's comment: In Fig.4, when the authors discussed OI effects on apoptosis and autophagy, the cell morphology changes are suggested to be described at the same time since they are intuitive, not only the cellular viability.*

Responses: We acknowledge your comments and suggestions very much, which are valuable in improving the quality of our manuscript. For this reason, we added relevant experiments to clarify the morphological and quantitative changes of chondrocytes in different groups (Fig. S1a-e). We added the description of the morphology of chondrocytes in the part of the results (p5, line126-131). Because of the COVID-19 pandemic is severe in part of China these days, C28/I2 chondrocyte cell line is unavailable lately, we used the rat primary chondrocytes instead. The details of the experiments are as follows:

S1: The morphological analysis of the chondrocytes in different groups (magnification, x200). a) The controls, the rat primary chondrocytes with no drugs

added were cultured for 48 h. b-e) Images of rat primary chondrocytes cultured with b) IL-1 β , c) IL-1 β +OI, d) CQ, e) CQ+OI for 48 h. f) Chondrocyte count among groups. Data represent the mean \pm SD (n=3), *, P<0.05.

2. **The reviewer's comment:** In Fig.6, the magnification times should be should be total objective magnification times, that is magnification times * eyepiece magnification times, not objective magnification.

Responses: Thank for your professional review on our manuscript. We changed the description of the magnification of the picture. (Page.23, Fig.6) The details of the changes are as follows:

3. **The reviewer's comment:** *The concentrations of IL-6, TNF-α et al should be easily and more accurately assayed and in the serum more in the synovial fluid. MMP3 and MMP13 may be detected using immunohistochemistry.*

Responses: Thank you very much for your comments. That is true the serum is easier to obtain than synovial fluid. However, we think that the inflammatory factors extracted from synovial fluid can better reflect the real situation of osteoarthritis in rats. Because the inflammatory factors secreted by chondrocytes reach the blood may reduce due to the degradation of lymph nodes. Support for this idea can be found in the literature¹. We also performed immunohistochemical detection of MMP13 in rats. The results indicated a trend consistent with that of ELISA (Fig.6). The details are as follows:

Fig. 6 OI induced autophagy marker expression and reduced inflammatory cytokine expression in rats with OA. **a-d)** Representative images of immunohistochemistry and the quantification of MMP13 and Beclin-1 expression (scale bar = 250-100 μ m). **e-h)** The concentrations of IL-6, TNF- α , MMP3 and MMP13 were evaluated by ELISA (n=3). The data are expressed by mean \pm SD. *, P<0.05, **, P<0.01, ***, P<0.001.

1. Peake, N., K. Khawaja, A. Myers, D. Jones, T. Cawston, A. Rowan, H. Foster, Levels of matrix metalloproteinase (MMP)-1 in paired sera and synovial fluids of juvenile idiopathic arthritis patients: relationship to inflammatory activity, MMP-3 and tissue inhibitor of metalloproteinases-1 in a longitudinal study. *Rheumatology* (Oxford, England), 2005. 44(11): p. 1383-9.

Reviewer #1 (Remarks to the Author):

The manuscript from Pan et al. describes the effect of cell permeable 4-octyl itaconate 4-OI on the human chondrocyte cell line C28/I2, and on an in vivo rat model of post-traumatic OA. The authors show that 4-OI reduces cell apoptosis and rescues proliferation of C28/I2 cells in response to IL-1B. Next, the authors show that markers of autophagy are increased while markers of autophagy inhibition are decreased. Furthermore, the authors show that 4-OI inhibited increases in PI3K/AKT/mTOR signaling in vitro. Next, the authors show that co-incubation of 4-OI with chloroquine, an autophagy inhibitor, reverses the anti-apoptotic and molecular effects seen by 4-OI. Finally, the authors inject 4-OI into joints of rats that received surgery to induce post-traumatic OA. The authors show that Beclin-1 is rescued and MMP-13 is reduced in vivo. Furthermore, markers of autophagy, apoptosis, cartilage degeneration and PI3K/AKT/mTOR signaling were modified in cartilage by 4-OI injection in vivo while circulating inflammatory markers were reduced.

Overall, the manuscript contains some interesting data indicating potential disease modifying effects of 4-OI for treatment of OA. However, there are some significant issues with the manuscript related to missing data and statistical analyses, conclusions being overstated, figure resolution and organization, and writing clarifications. Details of these issues are indicated below:

- 1) Line 69 – Can the authors clarify in the introduction what a mercaptan reaction is. This term should be defined and introduced here.
- 2) Throughout the results it is unclear, specifically for experiments using C28/I2 cells, how many independent experiments were performed and the number of technical replicates to create the mean from each independent experiment were used. Furthermore, for animal experiments, how many animals were used for each analysis? It is recommended that the authors change the data from plunger plots to dot plots to specifically detail each independent experiment/in vivo measure.

- 3) Line 80 – “moi” should be “M”
- 4) In a number of instances in the manuscript, when the authors discuss proliferation and apoptosis, there is no statistical analyses associated with these experiments, however, the authors indicate in the results writing that apoptosis was significantly inhibited and proliferation was rescued. Statistical analyses are needed to back-up these results. This includes All of figure 1 and panels in figure 4.
- 5) Line 92 – The title of this section is recommended to be amended. The data in Figure 2 only shows markers, not actual activity of autophagy and apoptosis, thus the section title should be changed accordingly.
- 6) In figure 2C, the pro-caspase-3 blot appears to be over saturated. Do the authors have a lower exposure of this blot to see differences in expression better? Similar situations appear to be the case in figures 4C&E.
- 7) Figure 4 and lines 124-153 - Figure 4 is completely out of order compared to what is described in the results. In addition, the blots and graphs in figure 4 are difficult to see as they are very small and cannot be accurately reviewed at this resolution.
- 8) Short-forms such as “p”, “t” and “CQ” should be listed in their long form at first instance in the manuscript.
- 9) Line 130 – can the authors clarify where the apoptosis data is in figure 4A? The data presented looks to be proliferation data.
- 10) Line 131 – the proliferation data in figure 4 needs statistical analysis.
- 11) Figure 4G – this data would benefit from some quantification as the authors indicate on line 141 that the fluorescence intensity was significantly enhanced, but without statistics to back this up.
- 12) Can the authors clarify if the zymogram activity was quantified in figure 4F? This needs to be done for appropriate statistical analysis.
- 13) In the methods, the authors describe their surgery as a DMM model. The surgery is actually a meniscectomy and not a DMM. Furthermore, the authors have not shown that the model induced OA, particularly cartilage

degeneration. OARSI scoring or similar scoring of the cartilage is needed to confirm OA was indeed induced in the model.

14) Figure 5A and lines 160-165 - These statements require quantification for MMP13 and Beclin. It is recommended that positive cells/total cells be calculated.

15) Lines 173-175 – Can the authors clarify where secretion data from cartilage tissue can be found? There is measures of these markers in the serum but secretion data from cartilage is missing.

16) Lines 194-196 – The data presented for in vitro studies is prophylactically cytoprotective. However, the in vivo results suggest that 4-OI attenuates disease after induction OA. What happens in culture when cells are treated with IL-1B first and followed by 4-OI? This would be a good experiment to support the in vivo finding using in vitro means to show that 4-OI could have therapeutic potential.

17) Line 206 - Chondrocytes do not transform to OA. It is recommended that the authors revise this statement.

18) Lines 222-224 – Can the authors clarify how fluorescence intensity of Beclin-1 more directly proved the ability of 4-OI to activate autophagy of chondrocytes?

19) Lines 243 - Although the molecular results are of interest, more attention needs to be paid to the in vivo results, as these results have more translatable potential vs. the in vitro results provided by the authors.

Reviewer #1 (Remarks to the Author):

The manuscript from Pan et al. describes the effect of cell permeable 4-octyl itaconate 4-OI on the human chondrocyte cell line C28/I2, and on an in vivo rat model of post-traumatic OA. The authors show that 4-OI reduces cell apoptosis and rescues proliferation of C28/I2 cells in response to IL-1B. Next, the authors show that markers of autophagy are increased while markers of autophagy inhibition are decreased. Furthermore, the authors show that 4-OI inhibited increases in PI3K/AKT/mTOR signaling in vitro. Next, the authors show that co-incubation of 4-OI with chloroquine, an autophagy inhibitor, reverses the anti-apoptotic and molecular effects seen by 4-OI. Finally, the authors inject 4-OI into joints of rats that received surgery to induce post-traumatic OA. The authors show that Beclin-1 is rescued and MMP-13 is reduced in vivo. Furthermore, markers of autophagy, apoptosis, cartilage degeneration and PI3K/AKT/mTOR signaling were modified in cartilage by 4-OI injection in vivo while circulating inflammatory markers were reduced.

Overall, the manuscript contains some interesting data indicating potential disease modifying effects of 4-OI for treatment of OA. However, there are some significant issues with the manuscript related to missing data and statistical analyses, conclusions being overstated, figure resolution and organization, and writing clarifications. Details of these issues are indicated below:

1. **The reviewer's comment:** Line 69 – Can the authors clarify in the introduction what a mercaptan reaction is. This term should be defined and introduced here.

69 (OI)[24]. OI and Itaconate have similar activity of mercaptan reaction, which makes
70 OI a suitable substitute for itaconate to study biological function. Itaconate has a

Responses: Thank for your professional review on our manuscript. The "mercaptan reaction" is a kind of chemical reaction and is of little concerns with this study. The description is deleted.

(Page.3, Line 66-67)

- The reviewer's comment:** Throughout the results it is unclear, specifically for experiments using C28/I2 cells, how many intended experiments were performed and the number of technical replicates to create the mean from each independent experiment were used. Furthermore, for animal experiments, how many animals were used for each analysis? It is recommended that the authors change the data from plunger plots to dot plots to specifically detail each independent experiment/in vivo measure.

Responses: Thank you very much for your comments. We changed the data from plunger plots to dot plots to specifically detail each independent experiment/in vivo measure. For details, please check the figure section (Fig.1-Fig.8).

- The reviewer's comment:** Line 80 – “moi” should be “M”.

Responses: We are very sorry to have such a problem, we have revised accordingly. (Page.3, Line76)

- The reviewer's comment:** In a number of instances in the manuscript, when the authors discuss proliferation and apoptosis, there is no statistical analyses associated with these experiments, however, the authors indicate in the results writing that apoptosis was significantly inhibited and proliferation was rescued. Statistical analyses are needed to back-up these results. This includes All of figure 1 and panels in figure 4.

Responses: Thank you very much for your comments. We have added the statistical analysis of Figure 1 and Figure 4. In addition, statistical analysis of all data was performed. (Fig.1-Fig.8).

Fig. 1 The effects of OI on C28/I2 cells **a)** The cytotoxic effects of OI on C28/I2 cells was measured using the MTT assay. The highest concentration at which OI was 100 μ M (n=3). The data are the mean \pm SD. *, P<0.05. **b)** Cell viability after treatment with OI for 48 h before exposure to 10 ng/mL IL-1 β for 24 h (n=3). The data are the mean \pm SD. *, P<0.05 vs. the Control group, #, P<0.05 vs. the IL-1 β group. **c-e)** Flow cytometry assay was performed to assess the number of dead cells (n=3).

Fig. 4 OI induced autophagy marker expression through the PI3K/AKT/mTOR signaling pathway. a) MTT assay showing the proliferation of C28/I2 cells treated with IL-1 β , OI and CQ (n=3). *, P<0.05, the CQ+OI group vs. the CQ group, the CQ group vs. the Control group; **b-d)** qRT-PCR was used to analyse the mRNA expression of LC3, Beclin-1, p62 in chondrocytes treated with IL-1 β , OI and CQ (n=3); **e-h)** WB was performed to analyse the expression levels of Beclin-1, LC3 and p62, which were normalized to GAPDH (n=3); **i-l)** Representative images of Western blotting analysis of the protein levels of p-AKT, T-AKT, p-PI3K, T-PI3K, p-mTOR and T-mTOR and the quantification of p-AKT/T-AKT, p-PI3K/T-PI3K, and p-mTOR/T-mTOR in the blots shown (n=3); **m-o)** The Western blot analysis of BAX, pro-caspase3 and cleaved-caspase3 protein expression and normalized to GAPDH (n=3). The data are expressed

by mean \pm SD. *, P<0.05, **, P<0.01, ***, P<0.001.

5. **The reviewer's comment:** Line 92 – The title of this section is recommended to be amended. The data in Figure 2 only shows markers, not actual activity of autophagy and apoptosis, thus the section title should be changed accordingly.

Responses: I am very grateful to your comments for the manuscript. In order to express the view of the article more accurately, we changed the section title to "OI promoted the expression of autophagy markers and inhibited the expression of apoptosis-related factors in C28/I2 cells". (Page.4, Line 91-92) At the same time, we have also changed the expression of other relevant parts of the article. (Page.5, Line 118-120, Line 143-144)

6. **The reviewer's comment:** In figure 2C, the pro-caspase-3 blot appears to be over saturated. Do the authors have a lower exposure of this blot to see differences in expression better? Similar situations appear to be the case in figures 4C&E.

Responses: Thanks for your reminder. The problem with the expression intensity of WB may be that our laboratory all used a slightly high intensity exposure when doing WB, which caused the WB results to be slightly over-exposed, here, we added a quantitative data to better understand the trend of proteins expression (Fig.1-Fig.8).

7. **The reviewer's comment:** Figure 4 and lines 124-153 - Figure 4 is completely out of order compared to what is described in the results. In addition, the blots and graphs in figure 4 are difficult to see as they are very small and cannot be accurately reviewed at this resolution.

Responses: We acknowledge your comments and suggestions very much,

which are valuable in improving the quality of our manuscript. We adjusted and split the figure 4 and changed the description in the results in order. (Page.5, Line120-141, Figure 4-5) The details of the changes are as follows:

Fig. 4 OI induced autophagy marker expression through the PI3K/AKT/mTOR signaling pathway. a) MTT assay showing the proliferation of C28/I2 cells treated with IL-1 β , OI and CQ (n=3). *, P<0.05, the CQ+OI group vs. the CQ group, the CQ group vs. the Control group; **b-d)** qRT-PCR was used to analyse the mRNA expression of LC3, Beclin-1, p62 in chondrocytes treated with IL-1 β , OI and CQ (n=3); **e-h)** WB was performed to analyse the expression levels of Beclin-1, LC3 and p62, which were normalized to GAPDH (n=3); **i-l)** Representative images of Western blotting analysis of the protein levels of p-AKT, T-AKT, p-PI3K, T-PI3K, p-mTOR and T-mTOR and the quantification of p-

AKT/T-AKT, p-PI3K/T-PI3K, and p-mTOR/T-mTOR in the blots shown (n=3); **m-o)** The Western blot analysis of BAX, pro-caspase3 and cleaved-caspase3 protein expression and normalized to GAPDH (n=3). The data are expressed by mean \pm SD. *, P<0.05, **, P<0.01, ***, P<0.001.

Fig. 5 Effects of OI on the expression of cartilage degradation-related proteins and autophagy markers. a-c) Zymographic analysis of the effects of IL-1 β , CQ and OI on the enzymatic activities of MMP3 and MMP13 and the quantification of MMP3 and MMP13 expression (n=3). **d-e)** Representative immunofluorescence photomicrograph of Beclin1 (red)-labelled chondrocytes and the fluorescence intensity quantification of Beclin1 expression (n=3) (scale bar = 50 μ m). Nuclei were stained with DAPI (blue). The data are expressed by mean \pm SD. *, P<0.05, **, P<0.01, ***, P<0.001.

8. **The reviewer's comment:** Short-forms such as "p", "t" and "CQ" should be listed in their long form at first instance in the manuscript.

Responses: Thank you very much for your comments. We have modified it accordingly. (Page.4, Line 110-112, Page.5, Line121). The details of the changes are as follows:

“Western blotting was performed to measure the protein expression of phosphorylated (p)-AKT, total (T)-AKT, phosphorylated (p)-PI3K, total (T)-PI3K, phosphorylated (p)-mTOR and total (T)-mTOR in C28/I2 cells in different groups (Fig. 3a).”

“we treated C28/I2 cells with chloroquine (CQ) (a specific inhibitor of autophagy) and compared the differences among the control, IL-1 β , IL-1 β +OI, CQ and CQ+OI groups.”

9. **The reviewer’s comment:** Line 130 – can the authors clarify where the apoptosis data is in figure 4A? The data presented looks to be proliferation data.

130 in Fig. 4A, CQ accelerated the apoptosis of chondrocytes, while the chondrocytes
131 growth inhibition effect of CQ was also significantly improved with the addition of
132 OI. Overall, this effect was better than that of the IL-1 β group. This suggested that OI

Responses: Thank you very much for your reminding. The data in figure 4A is real proliferation data. We made a writing error in previous version of the manuscript. The details of the changes are as follows:

“As shown in Fig. 4a, CQ slowed the proliferation of chondrocytes, while the inhibitory effect of CQ was markedly ameliorated by the addition of OI.”
(Page.5, Line 123-125)

10. **The reviewer’s comment:** Line 131 – the proliferation data in figure 4 needs statistical analysis.

Responses: We acknowledge your suggestion very much. We supplement the statistical analysis in figure 4a.

Fig.4a: MTT assay of the proliferation rate of chondrocytes treated with IL-1 β , OI and CQ (n=3). *, P<0.05 CQ+OI group vs CQ group, CQ group vs Control group.

11-12. The reviewer's comment: 11. Figure 4G – this data would benefit from some quantification as the authors indicate on line 141 that the fluorescence intensity was significantly enhanced, but without statistics to back this up.

12. Can the authors clarify if the zymogram activity was quantified in figure 4F? This needs to be done for appropriate statistical analysis.

Responses: We sincerely appreciate the valuable suggestions of the reviewer. The data of immunofluorescence and Zymographic were quantitatively and statistically analyzed.

Fig. 5 Effects of OI on the expression of cartilage degradation-related proteins and autophagy markers. a-c) Zymographic analysis of the effects of IL-1 β , CQ and OI on the enzymatic activities of MMP3 and MMP13 and the quantification of MMP3 and MMP13 expression (n=3). **d-e)** Representative immunofluorescence photomicrograph of Beclin1 (red)-labelled chondrocytes and the fluorescence intensity quantification of Beclin1 expression (n=3) (scale bar = 50 μ m). Nuclei were stained with DAPI (blue). The data are expressed by mean \pm SD. *, P<0.05, **, P<0.01, ***, P<0.001.

13. The reviewer's comment: In the methods, the authors describe their surgery as a DMM model. The surgery is actually a meniscectomy and not a DMM. Furthermore, the authors have not shown that the model induced OA, particularly cartilage degeneration. OARSI scoring or similar scoring of the cartilage is needed to confirm OA was indeed induced in the model.

Responses: We acknowledge the your comments and suggestions very much, which are valuable in improving the quality of our manuscript. We restated the surgical methods and added related references. (Page.12, Line 320-334) The OARSI scoring was employed to confirm the OA model. (Page.12, Line 333-334) The details of the changes are as follows:

“In this experiment, 5-week-old male Sprague Dawley rats (250–300 g) were anaesthetized with 2% pentobarbital sodium solution via intraperitoneal injection. The DMM model was used to simulate OA as previously described^{48, 49}. For the sham-operated controls, a similar incision was made on another leg of the rats, but the ligaments were left intact. The rats were allowed to recover for 4 weeks, and during this time they were allowed to move freely in the cages. The rats were randomly divided into three groups: the control group (sham-operated and treated with 100 μ L of normal saline on the first day of every week from the 4th to 12th week after surgery, n=6), the OA group (subjected to DMM, 100 μ L of normal saline treatment injected at the same as in the control group, n=6), and the OA+OI group (subjected to DMM, 100 μ L of OI (100 μ M) injected at the same as in the control group, n=6). At the end of the treatments, the animals were euthanized by an overdose of anaesthesia, and knee samples were harvested and fixed with 4% paraformaldehyde (PFA) for at least 48 h. Cartilage and synovial fluid were harvested for further analysis. The OARSI histopathology scoring system was used to evaluate OA severity (Fig. 8c).”

14. The reviewer’s comment: Figure 5A and lines 160-165 - These statements require quantification for MMP13 and Beclin1. It is recommended that positive cells/total cells be calculated.

Responses: We acknowledge your suggestion very much. We consider that MMP13 and Beclin1 are expressed more or less in the joint tissue of rats, so we used the intensity of expression for quantitative analysis. The details of the changes are as follows:

Fig. 6 OI induced autophagy marker expression and reduced inflammatory cytokine expression in rats with OA. a-d) Representative images of immunohistochemistry and the quantification of MMP13 and Beclin-1 expression (scale bar = 250-100 μ m). **e-h)** The concentrations of IL-6, TNF- α , MMP3 and MMP13 were evaluated by ELISA (n=3). The data are expressed by mean \pm SD. *, P<0.05, **, P<0.01, ***, P<0.001.

15. The reviewer's comment: Lines 173-175 – Can the authors clarify where secretion data from cartilage tissue can be found? There is measures of these markers in the serum but secretion data from cartilage is missing.

Responses: We sincerely appreciate the valuable suggestions. I'm sorry for our unclear description. Actually, the synovial fluid of rats was harvested for ELISA experiment. We have changed the description in the article. (Page.11, Line 301-302)

“The concentrations of IL-6, MMP3, MMP13 and TNF- α in the rat synovial fluid were determined by ELISA.”

16. The reviewer's comment: Lines 194-196 – The data presented for in vitro studies is prophylactically cytoprotective. However, the in vivo results suggest that 4-OI attenuates disease after induction OA. What happens in culture when cells are treated with IL-1 β first and followed by 4-OI? This would be a good experiment to support the in vivo finding using in vitro means to show that 4-OI could have therapeutic potential.

194 our study showed that the apoptosis of chondrocytes was reduced when chondrocytes
195 were pre-treated with OI before the addition of IL-1 β , which is another example of the
196 cytoprotective effect of OI. In our results, the toxic effect of different doses of OI on

Responses: We acknowledge the reviewer's comments and suggestions very much. The inflammatory degeneration of chondrocytes stimulated by IL-1 β is rapid, in order to avoid early rapid apoptosis caused by sudden stimulation of chondrocytes, we chose OI pretreatment. We believe that it can better reflect the real role of OI in chondrocytes. Similar behavior has been reported in literatures¹⁻³.

17. The reviewer's comment: Line 206 - Chondrocytes do not transform to OA. It is recommended that the authors revise this statement.

Responses: Thank you for reminding this mistake. We have revised accordingly. (Page.7, Line 198-199)

“which is closely related to the transformation of chondrocytes to OA chondrocytes.”

18. The reviewer's comment: Lines 222-224 – Can the authors clarify how

fluorescence intensity of Beclin-1 more directly proved the ability of 4-OI to activate autophagy of chondrocytes?

Responses: We sincerely appreciate the valuable suggestions of the reviewer. Our original manuscript did not accurately express the difference between autophagy markers and autophagy, we have corrected a number of inappropriate expressions on autophagy. (Page.4, Line 91-92; Page.5, Line 143-144, Line 130-132)

19. The reviewer's comment: Lines 243 - Although the molecular results are of interest, more attention needs to be paid to the in vivo results, as these results have more translatable potential vs. the in vitro results provided by the authors.

Responses: We acknowledge your suggestion very much. In the animal experiment part, we added relevant animal experiments to further illustrate the role of OI on alleviating OA. The details of the addition are as follows:

Fig. 8 OI improved osteoarthritis in OA rats. Histology of Control, AO and AO+OI rat knee model. The sections of knee joints were stained by a)

hematoxylin-eosin staining (scale bar = 50-100 μm) and **b**) safranin O/fast green staining (scale bar = 50-100 μm) after eight weeks of treatment. **c**) The OARSI score was evaluated based on safranin O/fast green staining. Data are means \pm SD (n=6). **, P<0.01, ****, P<0.0001.

About language and grammar: We carefully examined the details of the article, corrected the errors in the article, and modified the format according to the requirements of the journal.

References

1. Hua, L., F. Yu, X. Min, Y. Jian, G. Di, Four-octyl itaconate activates Keap1-Nrf2 signaling to protect neuronal cells from hydrogen peroxide. *Cell Communication and Signaling*, 2018. 16(1).
2. Linghu, K.-G., S.H. Xiong, G.D. Zhao, T. Zhang, W. Xiong, M. Zhao, X.-C. Shen, W. Xu, Z. Bian, Y. Wang, H. Yu, *Sigesbeckia orientalis* L. Extract Alleviated the Collagen Type II-Induced Arthritis Through Inhibiting Multi-Target-Mediated Synovial Hyperplasia and Inflammation. *Frontiers in pharmacology*, 2020. 11: p. 547913-547913.
3. Ohta, M., N. Chosa, S. Kyakumoto, S. Yokota, N. Okubo, A. Nemoto, M. Kamo, S. Joh, K. Satoh, A. Ishisaki, IL-1 β and TNF- α suppress TGF- β -promoted NGF expression in periodontal ligament-derived fibroblasts through inactivation of TGF- β -induced Smad2/3- and p38 MAPK-mediated signals. *International journal of molecular medicine*, 2018. 42(3): p. 1484-1494.

Reviewers' comments:

Reviewer #2 (Remarks to the Author):

This paper mainly investigated the effects of 4-octyl itaconate, a cell permeable itaconate derivative on the efficacy osteoarthritis in C2812 cells and DMM rat models. The authors found that OI can alleviate OA through inhibiting PI3K/AKT/mTOR pathway. This study has some academic values. The revised manuscript seems improved.

1. In Fig.4, when the authors discussed OI effects on apoptosis and autophagy, the cell morphology changes are suggested to be described at the same time since they are intuitive, not only the cellular viability .

2. In Fig.6, the magnification times should be total objective magnification times, that is magnification times * eyepiece magnification times, not objective magnification.

3. The concentrations of IL-6、 TNF- α et al should be easily and more accurately assayed and in the serum more in the synovial fluid. MMP3 and MMP13 may be detected using immunohistochemistry.

Response to Reviewers' comments

1. The reviewer's comment: In Fig.4, when the authors discussed OI effects on apoptosis and autophagy, the cell morphology changes are suggested to be described at the same time since they are intuitive, not only the cellular viability.

Responses: We acknowledge your comments and suggestions very much, which are valuable in improving the quality of our manuscript. For this reason, we added relevant experiments to clarify the morphological and quantitative changes of chondrocytes in different groups (Fig. S1a-e). We added the description of the morphology of chondrocytes in the part of the results (p5, line126-131). Because of the COVID-19 pandemic is severe in part of China these days, C28/I2 chondrocyte cell line is unavailable lately, we used the rat primary chondrocytes instead. The details of the experiments are as follows:

S1: The morphological analysis of the chondrocytes in different groups (magnification, x200). a) The controls, the rat primary chondrocytes with no drugs added were cultured for 48 h. b-e) Images of rat primary chondrocytes

cultured with b) IL-1 β , c) IL-1 β +OI, d) CQ, e) CQ+OI for 48 h. f) Chondrocyte count among groups. Data represent the mean \pm SD (n=3), *, P<0.05.

2. The reviewer's comment: In Fig.6, the magnification times should be should be total objective magnification times, that is magnification times * eyepiece magnification times, not objective magnification.

Responses: Thank for your professional review on our manuscript. We changed the description of the magnification of the picture. (Page.23, Fig.6)

The details of the changes are as follows:

3. The reviewer's comment: The concentrations of IL-6、TNF-α et al should be easily and more accurately assayed and in the serum more in the synovial fluid. MMP3 and MMP13 may be detected using immunohistochemistry.

Responses: Thank you very much for your comments. That is true the serum is easier to obtain than synovial fluid. However, we think that the inflammatory factors extracted from synovial fluid can better reflect the real situation of osteoarthritis in rats. Because the inflammatory factors secreted by chondrocytes reach the blood may reduce due to the degradation of lymph nodes. Support for this idea can be found in the literature¹. We also performed immunohistochemical detection of MMP13 in rats. The results

indicated a trend consistent with that of ELISA (Fig.6). The details are as follows:

Fig. 6 OI induced autophagy marker expression and reduced inflammatory cytokine expression in rats with OA. a-d) Representative images of immunohistochemistry and the quantification of MMP13 and Beclin-1 expression (scale bar = 250-100 μ m). **e-h)** The concentrations of IL-6, TNF- α , MMP3 and MMP13 were evaluated by ELISA (n=3). The data are expressed by mean \pm SD. *, P<0.05, **, P<0.01, ***, P<0.001.

1. Peake, N., K. Khawaja, A. Myers, D. Jones, T. Cawston, A. Rowan, H. Foster, Levels of matrix metalloproteinase (MMP)-1 in paired sera and synovial fluids of juvenile idiopathic arthritis patients: relationship to

inflammatory activity, MMP-3 and tissue inhibitor of metalloproteinases-1 in a longitudinal study. *Rheumatology (Oxford, England)*, 2005. 44(11): p. 1383-9.